# Caspase-1-driven neutrophil pyroptosis and its role in host susceptibility to *Pseudomonas aeruginosa*

Karin Santoni[1], David Pericat[1], Leana Gorse[1], Julien Buyck[2], Miriam Pinilla[1], Laure Prouvensier[2], Salimata Bagayoko[1], Audrey Hessel[1], Stephen Adonai Leon-Icaza[1], Elisabeth Bellard[1], Serge Mazères[1], Emilie Doz-Deblauwe[3], Nathalie Winter[3], Christophe Paget[4], Jean-Philippe Girard[1], Christine T. N. Pham[5], Céline Cougoule[1], Renaud Poincloux[1], Mohamed Lamkanfi[6], Emma Lefrançais[1], Etienne Meunier[1]¤‡*, Rémi Planès[1]‡*

1 Institute of Pharmacology and Structural Biology (IPBS), University of Toulouse, CNRS, Toulouse, France, 2 UFR Medicine and Pharmacy, INSERM U1070, University of Poitiers, Poitiers, France, 3 INRAE, Université de Tours, ISP, Nouzilly, France, 4 Institut national de la santé et de la recherche médicale, Centre d'Etude des Pathologies Respiratoires, UMR 1100, Tours, France, 5 Division of Rheumatology, Department of Internal Medicine, Washington University School of Medicine, Saint Louis, Missouri, United States of America, 6 Department of Internal Medicine and Pediatrics, Ghent University, Ghent, Belgium

¤ Current Address: Institute of Pharmacology and Structural Biology (IPBS), University of Toulouse, CNRS, Toulouse, France
‡ These authors equally supervised this work.
* etienne.meunier@ipbs.fr (EM); Remi.planes@ipbs.fr (RP)

**Data Availability Statement:** All relevant data are within the manuscript and its Supporting information files.

## Abstract

Multiple regulated neutrophil cell death programs contribute to host defense against infections. However, despite expressing all necessary inflammasome components, neutrophils are thought to be generally defective in Caspase-1-dependent pyroptosis. By screening different bacterial species, we found that several *Pseudomonas aeruginosa* (*P. aeruginosa*) strains trigger Caspase-1-dependent pyroptosis in human and murine neutrophils. Notably, deletion of Exotoxins U or S in *P. aeruginosa* enhanced neutrophil death to Caspase-1-dependent pyroptosis, suggesting that these exotoxins interfere with this pathway. Mechanistically, *P. aeruginosa* Flagellin activates the NLRC4 inflammasome, which supports Caspase-1-driven interleukin (IL)-1β secretion and Gasdermin D (GSDMD)-dependent neutrophil pyroptosis. Furthermore, *P. aeruginosa*-induced GSDMD activation triggers Calcium-dependent and Peptidyl Arginine Deaminase-4-driven histone citrullination and translocation of neutrophil DNA into the cell cytosol without inducing extracellular Neutrophil Extracellular Traps. Finally, we show that neutrophil Caspase-1 contributes to IL-1β production and susceptibility to pyroptosis-inducing *P. aeruginosa* strains *in vivo*. Overall, we demonstrate that neutrophils are not universally resistant for Caspase-1-dependent pyroptosis.

**Funding:** This project was supported by the Fonds de Recherche en Santé Respiratoire - Fondation du Souffle (to EL), ATIP-Avenir program (to EM), FRM "Amorçage Jeunes Equipes" (AJE20151034460 to EM) and the ERC (StG INFLAME 804249 to EM), the NIH (AR073752 to CTNP), the European Society of Clinical Microbiology and Infectious Diseases (ESCMID, to RP), Invivogen-CIFRE PhD grant (to MP), Invivogen post-doctoral fellowship (to RP) and a PhD fellowship from the Minister of Research of Mali and Campus France agency (to SB). The funders had no role in study design, data collection and analysis, decision to publish, or preparation of the manuscript.

**Competing interests:** The authors have declared that no competing interests exist.

## Author summary

Neutrophils play an essential role against infections. Although multiple neutrophil death programs contribute to host defense against infections, neutrophils are thought to be defective in Caspase-1-dependent pyroptosis. We screened several microbial species for the capacity to overcome neutrophil resistance to Caspase-1-driven pyroptosis, and show that the bacterium *Pseudomonas aeruginosa* specifically engages the NLRC4 inflammasome to promote Caspase-1-dependent Gasdermin D activation and subsequent neutrophil pyroptosis. Furthermore, NLRC4 inflammasome-driven pyroptosis leads to histone citrullination, nuclear DNA decondensation and expansion into the host cell cytosol. However, Neutrophil Extracellular Trap (NET) are not formed because DNA is kept in the intracellular space despite plasma membrane permeabilization and extracellular release of soluble and insoluble alarmins. Finally, *in vivo P. aeruginosa* infections highlight that Caspase-1-driven neutrophil pyroptosis is detrimental to the host upon *P. aeruginosa* infection. Altogether, our results demonstrate Caspase-1-dependent pyroptosis in neutrophils as a process that contributes to host susceptibility to *P. aeruginosa* infection.

## Introduction

Over the last 30 years, non-apoptotic forms of cell death have emerged as crucial processes driving inflammation, host defense against infections but also (auto) inflammatory pathologies [1]. NETosis is an antimicrobial and pro-inflammatory from of cell death in neutrophils that promotes the formation of extracellular web-like structures called Neutrophil Extracellular Traps (NETs) [2]. Although the importance of NETosis in host immunity to infections has been well established [2–5], NETosis dysregulation also associates to autoimmunity, host tissue damage, aberrant coagulation and thrombus formation, which all contribute to inflammatory pathologies such as sepsis and autoimmune lupus [6–13].

NETosis consists of sequential steps that start with nuclear envelope disintegration, DNA decondensation, cytosolic expansion of nuclear DNA and its subsequent expulsion through the plasma membrane [14]. Completion of DNA decondensation and expulsion requires various cellular effectors. Among them, neutrophil serine proteases (Neutrophil elastase, Cathepsin G, Proteinase 3) or Caspase-11 may cleave histones, which relaxes DNA tension [3, 13, 15–17]. In addition, granulocyte-enriched Protein arginine deaminase 4 (PAD4) citrullinates histone-bound DNA to neutralize arginine positive charges and facilitate nuclear DNA relaxation and decondensation [4, 18, 19]. In a third step, decondensed DNA physically binds neutrophil cytoplasmic granule factors such as Neutrophil Elastase (NE), Cathepsin G (CathG), Proteinase 3 (Pr3) and Myeloperoxidase (MPO) proteins [3, 15, 18]. Finally, sub-cortical actin network disassembly promotes efficient DNA extrusion through the permeabilized plasma membrane [18, 20].

Depending on the initial trigger, various signaling pathways such as calcium fluxes [17, 18], necroptosis-associated MLKL phosphorylation [21], ROS-induced Neutrophil protease release [15] or endotoxin-activated Caspase-11 [3, 5, 22] have been shown to induce NETosis. ROS- and Caspase-11-dependent NETosis have been shown to share the requirement for cleavage of the pyroptosis executioner Gasdermin D (GSDMD) by neutrophil serine proteases and Caspase-11, respectively [3, 16]. Active GSDMD forms pores on PIP2-enriched domains of the plasma and nuclear membranes of neutrophils, which ensures both IL-1ß secretion [23–26] and osmotic imbalance-induced DNA decondensation and expulsion [3, 16]. However, the link between ROS and Gasdermin-D-dependent NETosis requires more investigations as a

recent study could show that a described GSDMD inhibitor, the LDC559, is actually a ROS inhibitor, but not a GSDMD inhibitor [27].

Intriguingly, despite inducing GSDMD cleavage, neutrophils were reported to resist induction of Caspase-1-dependent pyroptosis upon NLRC4 inflammasome activation by *Salmonella* Typhimurium and *Burkholderia thaïlandensis*, or upon Nigericin/ATP-mediated NLRP3 inflammasome activation [3, 5, 28, 29]. However, recent studies indirectly challenged canonical pyroptosis impairment in neutrophils by showing that sterile activators, but also the SARS-CoV-2 virus, could also contribute to canonical NLRP3 inflammasome-dependent neutrophil death and subsequent NETosis [30, 31]. However, whether bacterial species exist that can induce neutrophil pyroptosis by canonical inflammasomes has remained an open question.

Lung infections by the bacterium *Pseudomonas aeruginosa* (*P. aeruginosa*) can promote acute or chronic, life-threatening infections in immunocompromised and hospitalized patients [32]. *P. aeruginosa* strains express a Type-3 Secretion System (T3SS) that allows injecting a specific set of virulence factors into host target cells, including macrophages and neutrophils [33]. T3SS-expressing *Pseudomonas aeruginosa* strains classically segregate into two mutually exclusive clades. Those expressing the bi-cistronic ADP-rybosylating and GTPase Activating Protein (GAP) virulence factor ExoS, and those expressing the lytic phospholipase of the patatin-like family, ExoU [33]. Common to most of *P. aeruginosa* strains is the expression of two other toxins, ExoY and ExoT, whose functions in bacterial infection still remain unclear. All Exo toxins are injected by the T3SS into host target cells upon infections. Finally *P. aeruginosa* strains also use their T3SS to inject Flagellin but also some of the T3SS components (needle) into host target cells, which promotes activation of the NAIP-NLRC4 inflammasome and subsequent Caspase-1-driven and GSDMD-dependent pyroptosis of macrophages [34–40]. Although numerous studies underlined that neutrophils are targeted by *Pseudomonas aeruginosa* virulence factors, which could promote NETosis [12, 41–43], the critical bacterial effector molecules and their host cell targets remain extensively debated. Intriguingly, defective expression of the enzyme NADPH oxidase (Nox2) sensitizes murine neutrophils to Caspase-1-driven neutrophil death upon infection with *Pseudomonas aeruginosa* [44], which suggests that under certain conditions neutrophils might be prone to undergo Caspase-1-dependent pyroptosis. Whether caspase-1-mediated pyroptosis also occurs in WT neutrophils, and what its putative molecular and immune significance might be remains unknown.

Here, we screened several bacterial species for their ability to bypass neutrophil resistance to canonical inflammasome-induced pyroptosis induction and found that the bacterial pathogen *Pseudomonas aeruginosa* triggers Caspase-1-dependent pyroptosis in human and murine neutrophils. Notably, deletion of Exotoxins U or S in *P. aeruginosa* entirely rewires neutrophil death towards Caspase-1-driven pyroptosis, suggesting that these bacterial Exotoxins somehow suppress caspase-1-mediated neutrophil pyroptosis. Mechanistically, *P. aeruginosa*-induced pyroptosis requires the expression of a functional Type-3 Secretion System (T3SS) and Flagellin, but not the T3SS-derived toxins ExoS, ExoT, ExoY or ExoU. Consequently, *P. aeruginosa* selectively activates the neutrophil NLRC4 inflammasome, which ensures Caspase-1-driven Gasdermin D (GSDMD) cleavage and the induction of neutrophil pyroptosis. Furthermore, we show that GSDMD activation promotes Calcium-dependent Peptidyl Arginine Deaminase 4 (PAD4) activation. PAD4 goes on to citrullinates histones, which leads to DNA decondensation and translocation of decondensed neutrophil DNA in the host cell cytosol without it being expulsed from the cells into the extracellular environment. Finally, we show by intravital microscopy that neutrophil pyroptosis occurs in lungs of P. aeruginosa-infected MRP8-GFP mice, and that neutrophil-targeted deletion of caspase-1 (in MRP8-CreCasp1^flox mice) reduces both IL-1β production and susceptibility to *P. aeruginosa* infection *in vivo*.

Overall, our results highlight that Caspase-1-dependent pyroptosis is a functional process in neutrophils that contributes to host susceptibility to *P. aeruginosa* infection *in vivo*.

## Results

### *Pseudomonas aeruginosa* strains variously trigger Caspase-1-dependent and -independent neutrophil lysis

To determine whether neutrophils may undergo pyroptosis upon bacterial infection, we infected WT and *Caspase-1*$^{-/-}$ (*Casp1*$^{-/-}$) mouse Bone Marrow Neutrophils (BMNs) with various bacterial strains that are known to activate different inflammasomes in macrophages, such as *Salmonella* Typhimurium (*S.* Typhimurium, strain SL1344), *Shigella flexnerii* (*S. flexnerii*, strain M90T), *Legionella pneumophilia* (*L. pneumophilia*, strain Philadelphia-1), *Burkholderia thailandensis* (*B. thailandensis*, strain E264), *Pseudomonas aeruginosa* (*P. aeruginosa*, strain PAO1), *Listeria* monocytogenes (*L. monocytogenes*, strain EGD), *Burkholderia cenocepaciae* (*B. cenocepaciae*, strain LMG 16656), *Francisella tularensis* spp *novicida* (*F. novicida*, strain U112), *Escherichia coli* (*E. Coli*, strain K12), *Staphylococcus aureus* (*S. aureus*, strain USA-300) and *Vibrio cholera* (*V. cholera*, strain El tor) (Fig 1A). In addition to tracking neutrophil lysis (LDH release), we measured IL-1β release as a hallmark of inflammasome activation. Given that neutrophils are short-lived cells that undergo spontaneous apoptosis over time, which can result in secondary necrosis and LDH release *in vitro*, we first determined a suitable timeframe in which neutrophils resist spontaneous lysis. Having determined that culture media of WT and *Casp1*$^{-/-}$ neutrophils lacked spontaneous LDH release during the first 4–5 hours (S1A Fig), we next performed infections within the first 3–4 hours in order to avoid confounding effects of spontaneous LDH release. With the notable exception of *Staphylococcus aureus*, we observed that all bacteria triggered Caspase-1-dependent IL-1β release in our experimental set-ups (Fig 1A), thus confirming Caspase-1 activation in neutrophils upon infection with various bacterial pathogens. In addition, despite most of the tested bacteria inducing significant neutrophil lysis, only LDH release by *Pseudomonas aeruginosa* (PAO1 strain) was partially dependent on Caspase-1 (further referred to as pyroptosis) (Fig 1A). Furthermore, we confirmed that infection of human blood neutrophils with two different *Pseudomonas aeruginosa* strains (PAO1 and CHA) also triggered a Caspase-1-dependent, yet Caspase-4/5, -3/7 and -8-independent, IL-1ß release and neutrophil lysis (Fig 1B). These results demonstrate that various bacteria can trigger neutrophil lysis in a Caspase-1-independent manner, and that *Pseudomonas aeruginosa* PAO1 strain specifically promotes neutrophil lysis through a mechanism that is partially Caspase-1-dependent.

*P. aeruginosa* T3SS-mediated injection of Flagellin and toxins (ExoS, U, T, Y) play a major role in its virulence. To determine the role of these virulence factors in *P. aeruginosa*-induced neutrophil pyroptosis, we infected murine BMNs with WT *P. aeruginosa* PAO1 or mutant strains that are deficient in expression of individual toxins (PAO1$^{\Delta ExoS}$, PAO1$^{\Delta ExoT}$, PAO1$^{\Delta ExoY}$); deficient for the expression of Flagellin (PAO1$^{\Delta FliC}$); or for the T3SS (PAO1$^{\Delta ExsA}$) (Figs 1C and S1B). We parallelly measured the ability of WT and *Casp1*$^{-/-}$ neutrophils infected with these strains to undergo cell lysis (LDH release), to promote IL1β release, and to exhibit plasma membrane permeabilization (SYTOX Green incorporation) (Figs 1C and S1B). Our results show that *P. aeruginosa* strains lacking expression of T3SS or Flagellin were unable to promote robust Caspase-1-dependent neutrophil lysis, IL-1β release and plasma membrane permeabilization, suggesting that T3SS and Flagellin are major effectors of Caspase-1-driven neutrophil death (Figs S1B and 1C). Flagellin deficiency modifies the physical recognition and phagocytosis of *P. aeruginosa* by macrophages and neutrophils [45, 46] (S1C Fig), which could indirectly alter the amount of Flagellin injected into host target cells. To ensure similar

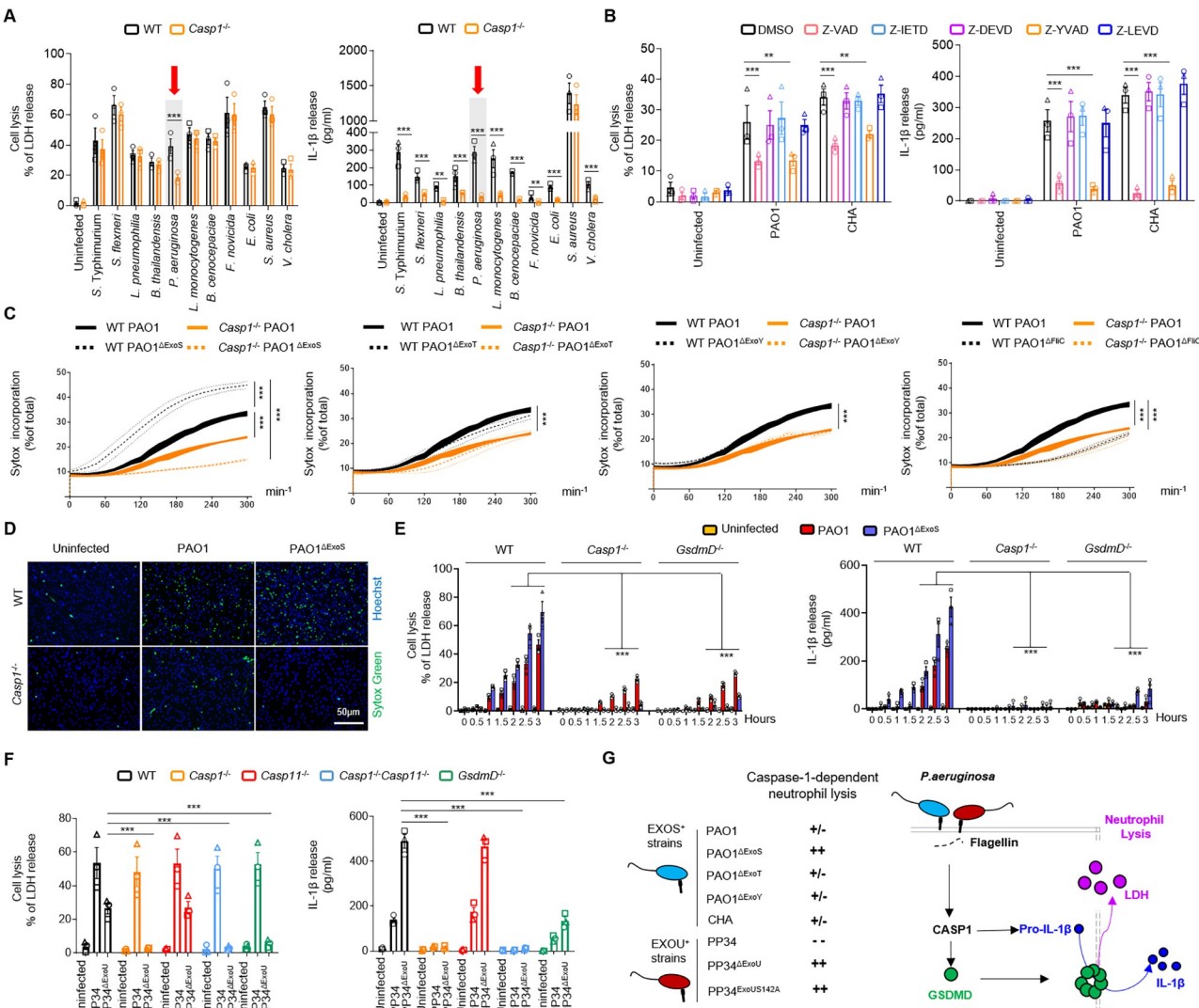

**Fig 1. Various *Pseudomonas aeruginosa* strains trigger Caspase-1-dependent and -independent neutrophil lysis. A.** Measure of cell lysis (release of LDH) and IL-1β release in WT or *Casp1*[-/-] murine Bone Marrow Neutrophils (BMNs) infected for 3 hours with various bacteria at a multiplicity of infection (MOI) of 10. **p ≤ 0.01 ***p ≤ 0.001, Two-Way Anova with multiple comparisons. Values are expressed as mean ± SEM. Graphs show combined values from three independent experiments. **B.** Measure of cell lysis (release of LDH) and IL-1β release in human blood neutrophils infected for 3 hours with *Pseudomonas aeruginosa* strains PAO1 or CHA (MOI 5) in presence/absence of various Caspase inhibitors, Z-VAD (pan Caspase, 20μM), Z-YVAD (Casp1 inhibitor, 40μM), Z-DEVD (Casp3 inhibitor, 40μM), Z-IETD (Casp8 inhibitor, 40μM) or Z-LEVD (Casp4/5 inhibitor, 40μM). **p ≤ 0.01 ***p ≤ 0.001, Two-Way Anova with multiple comparisons. Values are expressed as mean ± SEM. Graphs show values from three independent experiments with one different donor each time. **C, D.** Measure of plasma membrane permeabilization and associated fluorescent microscopy images over time using SYTOX Green incorporation in WT or *Casp1*[-/-] BMNs infected with *Pseudomonas aeruginosa* PAO1 or various isogenic mutants lacking Flagellin (FliC, PAO1[ΔFliC]) or T3SS-derived toxins ExoS, ExoY, ExoT (PAO1[ΔExoS], PAO1[ΔExoY], PAO1[ΔExoT]). ***p ≤ 0.001, Two-Way Anova with multiple comparisons. Values are expressed as mean ± SEM. Graphs show combined values from three independent experiments. Hoechst (Blue, DNA), Sytox Green (DNA from neutrophils with permeabilized membranes, Green). Scale Bar: 50μm. **E.** Measure of cell lysis (release of LDH) and IL-1β release in WT, *Casp1*[-/-] and *GsdmD*[-/-] murine Bone Marrow Neutrophils (BMNs) infected for the indicated times with PAO1 or its isogenic mutant PAO1[ΔExoS] at an MOI of 10. ***p ≤ 0.001, Two-Way Anova with multiple comparisons. Values are expressed as mean ± SEM. Graphs show combined values from three independent experiments. **F.** Measure of cell lysis (release of LDH) and IL-1β release in WT, *Casp1*[-/-], *Casp11*[-/-], *Casp1*[-/-]*Casp11*[-/-] and *GsdmD*[-/-] murine Bone Marrow Neutrophils (BMNs) infected for 3 hours with *Pseudomonas aeruginosa* PP34 strain or its isogenic mutant PP34[ΔExoU] at a multiplicity of infection (MOI) of 2. ***p ≤ 0.001, Two-Way Anova with multiple comparisons. Values are expressed as mean ± SEM. Graphs show combined values from three independent experiments. **G.** Overview of the importance of Caspase-1 at driving neutrophil pyroptosis in response to various *Pseudomonas aeruginosa* strains.

phagocytosis/recognition of *Pseudomonas aeruginosa* strains by neutrophils, we genetically invalidated the expression of Flagellin motors (namely MotABCD) in a WT PAO1 and Flagellin-deficient (PAO1$^{\Delta FliC}$) genetic background. Both PAO1$^{\Delta MotABCD}$ and PAO1$^{\Delta MotABCD/\Delta FliC}$ showed similar neutrophil uptake after 45 minutes of infection (S1C Fig), which allowed evaluating the direct importance of Flagellin in *P. aeruginosa*-driven Caspase-1-dependent neutrophil pyroptosis. Kinetic analysis of neutrophil lysis (LDH release) and IL-1β production in WT and *Casp1*$^{-/-}$ neutrophils showed that PAO1$^{\Delta MotABCD}$, but not PAO1$^{\Delta MotABCD/\Delta FliC}$, induced Caspase-1-dependent neutrophil lysis and IL-1β release over time (S1D Fig), suggesting that Flagellin plays a major role in *P. aeruginosa*-induced neutrophil pyroptosis.

To the contrary, deletion of Exo-Y or Exo-T in *P. aeruginosa* PAO1 did not significantly influence neutrophil lysis, IL-1β release or plasma membrane permeabilization (Fig 1C and 1D, and S1B). However, infection of neutrophils with *P. aeruginosa* PAO1 deficient for ExoS triggered significantly increased lysis, membrane permabilization and IL-1β release (Fig 1C and 1D, and S1B). In addition, PAO1$^{\Delta ExoS}$-induced neutrophil lysis was strongly reduced in *Caspase-1*-deficient neutrophils, suggesting that ExoS expression suppresses Caspase-1-dependent neutrophil death (Fig 1C and 1D, and S1B).

Caspase-1 cleavage of the pyroptosis executioner Gasdermin D (GSDMD) is the central effector mechanism of canonical inflammasome-induced pyroptosis in macrophages. Kinetic measurements confirmed that both *Casp1*$^{-/-}$ and *GsdmD*$^{-/-}$ neutrophils exhibited reduced LDH and IL-1β release upon *P. aeruginosa* infection (Fig 1E). Consistent with our earlier observations (Fig 1C and 1D, and S1B), neutrophil lysis and IL-1β release by the PAO1$^{\Delta ExoS}$ mutant (and to a lower extent also PAO1-induced responses) were strongly dependent on CASP1 and GSDMD expression (Fig 1E).

*P. aeruginosa* strains PAO1 and CHA belong to the 70% of the *P. aeruginosa* strains that express the Exotoxin S (ExoS). As 30% of *P. aeruginosa* strains do not express the Exotoxin S (ExoS) but instead are characterized as expressing the extremely lytic phospholipase toxin ExoU, we expanded our analysis to neutrophils infected with a strain expressing a catalytically inactive mutant of ExoU (PP34$^{ExoUS142A}$). In addition, we examined the response of neutrophils to a *P. aeruginosa* strain that naturally expresses ExoU (PP34 strain) or a mutant thereof that is genetically deficient for ExoU (PP34$^{\Delta ExoU}$). These experiments showed that the *P. aeruginosa* PP34 strain induces neutrophil lysis, yet in a Caspase-1- and Gasdermin D-independent manner (Figs 1F and S1E). However, both human and murine neutrophils infected with PP34$^{\Delta ExoU}$ or PP34$^{ExoUS142A}$ showed robust Caspase-1-dependent neutrophil lysis, plasma membrane permeabilization and IL-1β release (Figs 1F, S1E and S1F). These results suggest that Caspase-1 and GSDMD-dependent pyroptosis is masked or suppressed by ExoU activity in the *P. aeruginosa* PP34 strain.

Altogether, our results show the unexpected ability of Caspase-1 to promote GSDMD-dependent neutrophil pyroptosis upon *Pseudomonas aeruginosa* infection (Fig 1G).

## NLRC4 drives *P. aeruginosa*-induced neutrophil pyroptosis

Next, we investigated the upstream molecular pathways by which *P. aeruginosa* promoted Caspase-1-dependent neutrophil pyroptosis. To address this specific question, we used the most potent pyroptotic strains of *P. aeruginosa*, namely PAO1$^{\Delta ExoS}$, PP34$^{\Delta ExoU}$ or PP34$^{ExoUS142A}$ (further referred to as "pyroptotic strains" for clarity) (Fig 1G). We infected WT murine neutrophils or neutrophils from mice lacking expression of various inflammasome sensors, namely *Casp1*$^{-/-}$, *Casp11*$^{-/-}$, *Casp1*$^{-/-}$*Casp11*$^{-/-}$, *Nlrp3*$^{-/-}$, *AIM2*$^{-/—}$, *Nlrc4*$^{-/-}$ and *ASC*$^{-/-}$ (S2A Fig). Among the different tested neutrophil genotypes, significant resistance to pyroptotic cell lysis (LDH release) was only observed in *Casp1*$^{-/-}$, *Casp1*$^{-/-}$*Casp11*$^{-/-}$, *Nlrc4*$^{-/-}$ and *ASC*$^{-/-}$ BMNs

upon infection with *P. aeruginosa* pyroptotic strains PAO1$^{\Delta ExoS}$ and PP34$^{\Delta ExoU}$ (S2A Fig). Contrastingly, *Casp11*$^{-/-}$, *Nlrp3*$^{-/-}$ and *AIM2*$^{-/-}$ neutrophils exhibited similar lysis levels (LDH release) as observed in WT BMNs (S2A Fig). This suggests that NLRC4—but not NLRP3, Caspase-11 or AIM2—efficiently promotes neutrophil pyroptosis upon infection with *P. aeruginosa* pyroptotic strains (S2A Fig). The role of NLRC4 in *P. aeruginosa*-induced neutrophil cell death is confined to Caspase-1-dependent pyroptosis because infection of WT and *Nlrc4*$^{-/-}$ BMNs with *P. aeruginosa* strains that trigger Caspase-1-independent neutrophil lysis (PP34) resulted in similar LDH release levels, akin to our earlier results in *Casp1*$^{-/-}$ and *GsdmD*$^{-/-}$ neutrophils (Figs 1F and 2A). In contrast, infection of WT and *Nlrc4*$^{-/-}$ BMNs with *P. aeruginosa* strains of which neutrophil cell lysis is partially Caspase-1-dependent (PAO1 and CHA strains) also showed a partial involvement of NLRC4 in controlling neutrophil lysis (Fig 2A). Importantly, IL-1β release was entirely dependent on NLRC4 upon infection with any of these different *Pseudomonas aeruginosa* strains, whereas the pyroptotic strains PAO1$^{\Delta ExoS}$, CHA$^{\Delta ExoS}$ and PP34$^{\Delta ExoU}$ triggered neutrophil lysis and IL-1β release that was fully NLRC4-dependent (Fig 2A).

Further analysis of Caspase-1 (p20) and GSDMD (p30) processing in response to various *P. aeruginosa* strains showed that the pyroptotic strains PAO1$^{\Delta ExoS}$, PP34$^{ExoUS142A}$ and PP34$^{\Delta ExoU}$ as well as partially Caspase-1-dependent PAO1 strain triggered robust neutrophil Caspase-1 and GSDMD processing, a process that required NLRC4 expression (Fig 2B and 2C). As expected, a PAO1 mutant strain lacking the T3SS regulator ExsA (PAO1$^{\Delta ExsA}$) failed to induce robust Caspase-1 and GSDMD processing (Fig 2B). Finally, the PP34 strain, which promotes NLRC4- and Caspase-1-independent neutrophil lysis, also failed to trigger efficient cleavage of Caspase-1 and GSDMD, suggesting that PP34-induced neutrophil lysis is inflammasome-independent (Fig 2C). These results suggest that NLRC4 drives Caspase-1 and GSDMD cleavage in response to various *P. aeruginosa* strains that are capable of inducing neutrophil pyroptosis, namely PAO1, PAO1$^{\Delta ExoS}$, CHA, CHA$^{\Delta ExoS}$, PP34$^{\Delta ExoU}$ and PP34$^{ExoUS142A}$, but not in response to the PP34 strain that induces inflammasome-independent neutrophil lysis.

Based on this information, we sought to determine whether *P. aeruginosa* infection also induced neutrophil NLRC4 inflammasome activation *in vivo*. To this end, we infected ASC-Citrine mice with low doses (1.10$^5$ CFUs) of *P. aeruginosa* strains that specifically triggered NLRC4-dependent neutrophil pyroptosis, namely PP34$^{\Delta ExoU}$. As control, we included its isogenic mutant PP34$^{\Delta ExoU/\Delta FliC}$ that is deficient in Flagellin expression and hence unable to trigger NLRC4-dependent neutrophil pyroptosis *in vitro* (Figs 2D and S2B). ImageStreamX analysis of neutrophils presenting an active ASC supramolecular speck (ASC speck$^+$/LY6G$^+$ neutrophils) showed that PP34$^{\Delta ExoU}$ infection triggered inflammasome activation in neutrophils, and that the amount of ASC speck$^+$ neutrophils was reduced when mice where infected with Flagellin-deficient PP34$^{\Delta ExoU/\Delta FliC}$ mutant bacteria (Figs 2D and S2B). Altogether, these results show that the NLRC4/CASP1/GSDMD axis is fully functional to promote neutrophil pyroptosis in response to several *P. aeruginosa* strains, namely PAO1, PAO1$^{\Delta ExoS}$, CHA, CHA$^{\Delta ExoS}$, PP34$^{\Delta ExoU}$ or PP34$^{ExoUS142A}$ (Fig 2E).

## *P. aeruginosa*-induced inflammasome activation induces neutrophil DNA decondensation

As multiple cell death pathways such as Caspase-11-induced pyroptosis, MLKL-driven necroptosis, Mitogen- and Calcium-induced NETosis, have been linked to a direct or secondary induction of neutrophil DNA decondensation and release, a hallmark of NETosis, we next sought to determine whether *P. aeruginosa*-induced neutrophil inflammasome activation

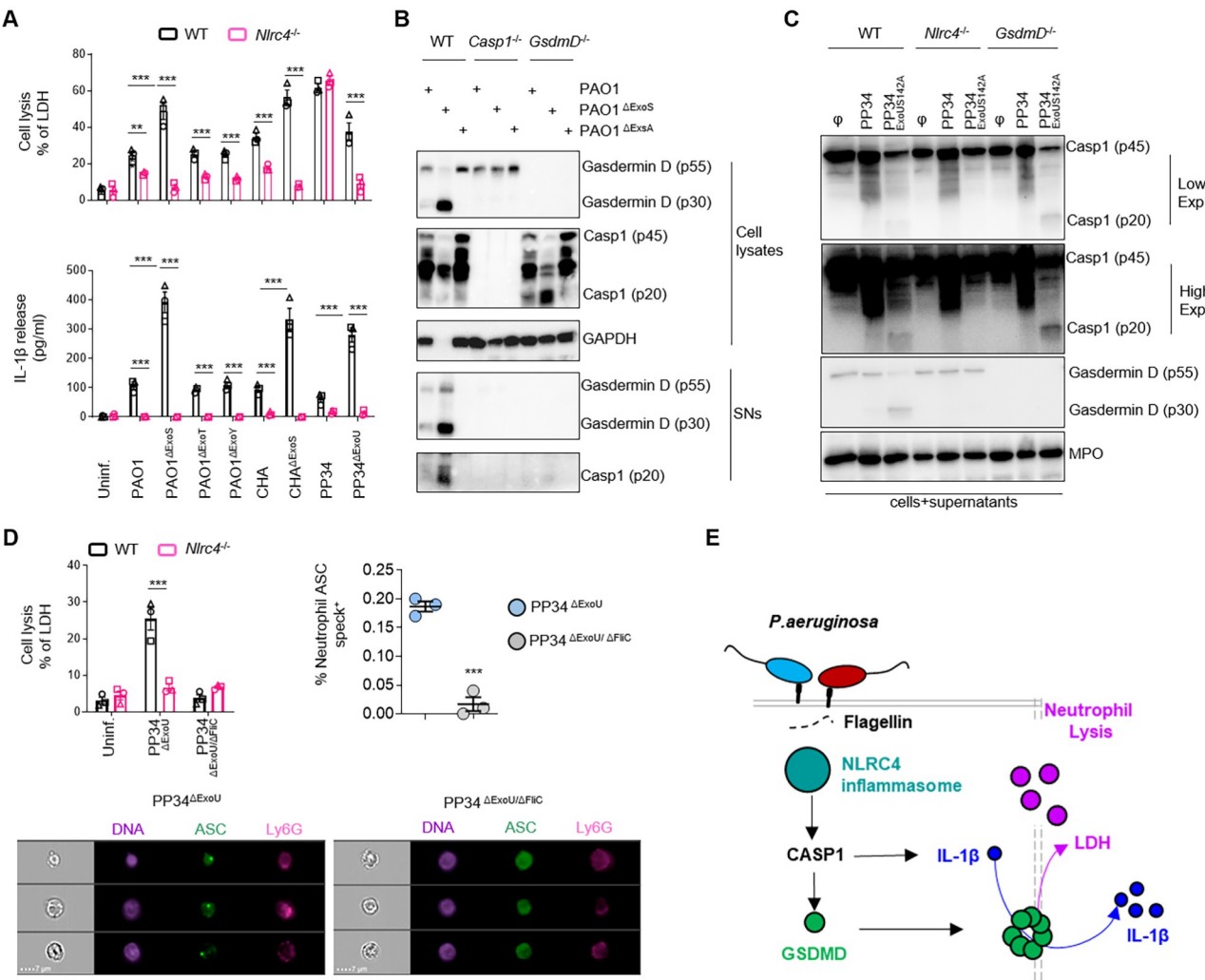

**Fig 2. _P. aeruginosa_ infection engages a canonical NLRC4-Caspase-1-Gasdermin D-dependent pyroptosis axis in neutrophils. A.** Measure of cell lysis (release of LDH) and IL-1β release in WT or _Nlrc4$^{-/-}$_ murine Bone Marrow Neutrophils (BMNs) infected for 3 hours with _Pseudomonas aeruginosa_ PAO1, CHA and PP34 strains and their isogenic mutants lacking T3SS-derived toxins ExoS, ExoY, ExoT or ExoU) at a multiplicity of infection (MOI) of 10 (PAO1 and CHA strains) and 2 (PP34 strains). $^{**}p \leq 0.01$ $^{***}p \leq 0.001$, Two-Way Anova with multiple comparisons. Values are expressed as mean ± SEM. Graphs show combined values from three independent experiments. **B.** Immunoblotting of GAPDH, pro-forms of Caspase-1 (p45) and Gasdermin-D (p55), processed Caspase-1 (p20) and Gasdermin D (p30), in WT, _Casp1$^{-/-}$_ and _GsdmD$^{-/-}$_ in cell lysates and cell supernatants (SNs) of BMNs infected for 3 hours with PAO1 or its isogenic mutants lacking T3SS expression (PAO1$^{\Delta ExsA}$) or ExoS (PAO1$^{\Delta ExoS}$) at a multiplicity of infection (MOI) of 10. Immunoblots show lysates and supernatants from one experiment performed three times. **C.** Immunoblotting of Myeloperoxidase (MPO), pro-forms of Caspase-1 (p45) and Gasdermin-D (p55), processed Caspase-1 (p20) and Gasdermin D (p30), in WT, _Nlrc4$^{-/-}$_ and _GsdmD$^{-/-}$_ BMNs infected for 3 hours with _P. aeruginosa_ strain PP34 or its isogenic mutant lacking ExoU activity (PP34$^{ExoUS142A}$) at a multiplicity of infection (MOI) of 2. Immunoblots show combined lysates and supernatants from one experiment performed three times. **D.** Measure of cell lysis (release of LDH) in WT and _Nlrc4$^{-/-}$_ BMNs infected for 3 hours with _P. aeruginosa_ mutant strains PP34$^{\Delta ExoU}$ or PP34$^{\Delta ExoU/\Delta FliC}$ at a multiplicity of infection (MOI) of 2 and Imagestream experiments and quantifications of _in vivo_ formation of ASC specks in bronchoalveolar (BALs) neutrophils from ASC-Citrine mice intranasally infected with 1.10$^5$ PP34$^{\Delta ExoU}$ or PP34$^{\Delta ExoU/\Delta FliC}$ for 6 hours. The gating strategy used to evaluate inflammasome activation in neutrophils was performed as follow: (i) a gate was set on cells in focus [Cells in Focus] and (ii) a sub-gate was created on single cells [Single Cells]. Then we gated first on (iii) LY6G+ Neutrophils [LY6G+] and second on (iv) ASC-citrine+ and Hoechst+ cells [Hoechst+/ASC-Citrine+] within LY6G+ population. (v) To distinguish cells with active (ASC-speck) versus inactive inflammasome (Diffuse ASC), we plotted the Intensity with the area of ASC-citrine. This strategy allows to distinguish cells with active inflammasome that were visualized and quantified. For Imagestream experiments $^{***}p \leq 0.001$, T-test with Bonferroni correction. Values are expressed as mean ± SEM. Graphs show one experiment representative of two independent experiments. For neutrophil _in vitro_ experiments, $^{***}p \leq 0.001$, Two-Way Anova with multiple comparisons. Values are expressed as mean ± SEM. Graphs show combined values from three independent experiments. **E.** Overview of the importance of the NLRC4 inflammasome at driving neutrophil pyroptosis in response to various _Pseudomonas aeruginosa_ strains.

could also lead to DNA decondensation and expulsion. We infected WT, $Casp1^{-/-}$ or $GsdmD^{-/-}$ murine BMNs with the pyroptotic *P. aeruginosa* strains PAO1, PAO1$^{\Delta ExoS}$, PP34$^{ExoUS142A}$, or with the PP34 strain that triggers Caspase-1-independent neutrophil lysis (Figs 1 and 2). We specifically monitored the presence of DNA Neutrophil Extracellular Traps (NETs) using Scanning Electron Microscopy (SEM) (Fig 3A). PP34, and to a lower extend PAO1, induced NETs in WT, $Casp1^{-/-}$ and $GsdmD^{-/-}$ neutrophils (Fig 3A). However, we observed that the fully pyroptotic strains PAO1$^{\Delta ExoS}$ and PP34$^{ExoUS142A}$ failed to induce NETs (Fig 3A), which suggests that Caspase-1-induced neutrophil pyroptosis does not promote NETosis.

Rather, immunofluorescence experiments of WT, $Casp1^{-/-}$ and $GsdmD^{-/-}$ neutrophils infected with the fully pyroptotic *P. aeruginosa* strain PP34$^{ExoUS142A}$ showed efficient DNA decondensation as well as exit from the nuclear envelope (Lamin-B1 staining) but little or no extracellular DNA release from BMNs exhibiting an active inflammasome complex (referred as ASC specks, ASC$^+$), a process also observed in primary human blood neutrophils (Fig 3A and 3B, and S3A). Further experiments using time lapse fluorescent microscopy on ASC-Citrine neutrophils infected with the pyroptotic strain PP34$^{ExoUS142A}$ or the NETosis-inducing strain PP34 showed that both bacterial strains triggered efficient neutrophil DNA decondensation (Fig 3C and S1 and S2 Movies). However, pyroptotic neutrophils uniquely failed to complete DNA release out from the plasma membrane (stained with WGA) (Fig 3C). These observations were also confirmed using 3D reconstruction of confocal fluorescent images (S3 and S4 Movies). Specifically, analyzing DNA decondensation induced by the pyroptotic strain PP34$^{ExoUS142A}$ or the NETotic strain PP34 in ASC-Citrine neutrophils, we observed that in ASC specks$^+$ cells, DNA efficiently decondensed and filled the entire intracellular space, but did not cross the plasma membrane (stained with WGA) (Fig 3C and S3 and S4 Movies). Given that NETosis features histone-bound DNA complexes outside from neutrophils, we reasoned that during pyroptosis, neutrophils might keep histone-bound DNA trapped intracellularly. Consistently, immunoblotting of histones in various neutrophil fractions (soluble, insoluble and supernatant) showed that PP34-induced NETosis efficiently promoted histone release in the extracellular medium (S3B Fig). However, the pyroptotic PP34$^{ExoUS142A}$ strain failed to induce extracellular histone release, although it efficiently promoted the release of intracellular soluble and insoluble components such as GAPDH, the nuclear membrane structural component Lamin B1, the nuclear alarmin HMGB1 or NLRC4 in the extracellular environment (S3B Fig). Importantly, this process required NLRC4 expression (S3B Fig). This suggests that NLRC4-dependent neutrophil pyroptosis specifically promotes DNA decondensation and release from the nuclear membrane while retaining histone-bound DNA intracellularly despite soluble and insoluble factors being released in the extracellular environment (S3C Fig).

## Inflammasome-induced neutrophil DNA decondensation is driven by Calcium-activated PAD4 downstream of GSDMD

Next, we wondered about the mechanisms by which inflammasome signaling promotes DNA decondensation and cytosolic redistribution. Two mechanisms that promote DNA decondensation in neutrophils have been reported. First, Neutrophil Serine proteases Neutrophil Elastase (NE), Cathepsin G (CatG) and Proteinase 3 (Pr3) can cleave Histones, which will favor DNA decondensation [15]. Second, upon Calcium release, the enzyme PAD4 promotes Histone Citrullination and subsequent DNA decondensation in neutrophils [4, 6, 7, 47].

We first explored if Caspase-1-induced neutrophil DNA delobulation and release required Neutrophil Serine proteases. We examined the extent of DNA decondensation in WT neutrophils and neutrophils lacking the three proteases NE, CatG and Pr3 ($NE^{-/-}CatG^{-/-}Pr3^{-/-}$) upon

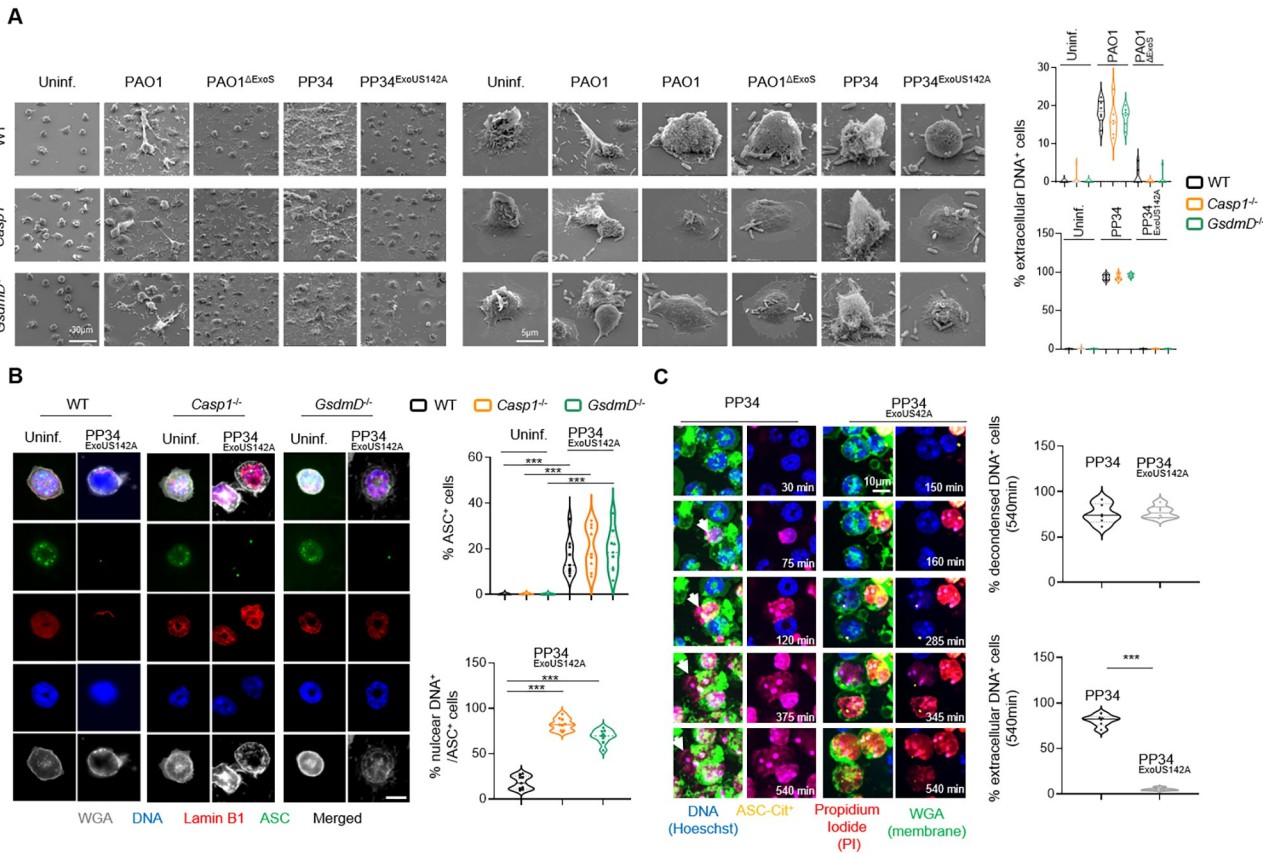

**Fig 3. NLRC4 inflammasome signaling in neutrophils promotes both pyroptosis and DNA decondensation but not DNA expulsion. A.** Scanning electron microscopy (SEM) observations and quantifications of pyroptosis in WT, *Casp1⁻/⁻* and *GsdmD⁻/⁻* BMNs 3 hours after infection with *P. aeruginosa* strains PAO1, PAO1^ΔExoS, PP34 and PP34^ExoUS142A at an MOI of 10 (PAO1 strains) and 2 (PP34 strains). Images are representative of one experiment performed 3 times. Scale bars are directly indicated in the figure. For quantifications, the percentage of cells exhibiting extracellular DNA was determined by quantifying the ratios of cells positives for extracellular DNA over the total of cells. 6–10 fields from n = 3 independent experiments were analyzed. Values are expressed as mean ± SEM. ***p ≤ 0.001, Two-Way Anova with multiple comparisons. **B.** Confocal microscopy observations and quantifications of WT, *Casp1⁻/⁻* and *GsdmD⁻/⁻* BMNs infected for 3 hours with the pyroptotic strain PP34^ExoUS142A (MOI 2) and harboring ASC complexes, decondensed DNA and nuclear membrane (LaminB1). Nucleus (blue) was stained with Hoechst; LaminB1 is in red (anti LaminB1); ASC is in Green (anti-ASC); plasma membrane is in grey (WGA staining). Scale bar 10µm. Images are representative of one experiment performed three times. For quantifications, the percentage of cells with ASC complexes and nuclear DNA was determined by quantifying the ratios of cells positives for ASC speckles and nuclear DNA. 6–10 fields from n = 3 independent experiments were analyzed. Values are expressed as mean ± SEM. ***p ≤ 0.001, Two-Way Anova with multiple comparisons. **C.** Representative time lapse fluorescence microscopy images and quantifications of ASC-Citrine murine BMNs infected with the NETotic strain PP34 or the pyroptotic strain PP34^ExoUS142A (MOI 2) for 9 hours (540 minutes). Nucleus (blue) was stained with Hoechst; ASC is in yellow (ASC-Citrine); plasma membrane is in green (WGA staining); plasma membrane permeabilization is stained in red (cell impermanent DNA dye Propidium Iodide, PI). Images are representative of one movie out of three independent movies. For quantifications, the percentage of cells harboring decondensed DNA and/or extracellular decondensed DNA was determined by quantifying the ratios of cells with decondensed DNA (area surface) or cells with decondensed DNA crossing WGA staining (DNA area surface outside from plasma membrane) over the total amount of cells. At least 6 fields containing each 20–30 cells were quantified. Scale bar 10µm. ***p ≤ 0.001, T-test with Bonferroni correction.

infection with the fully pyroptotic strain PP34^ExoUS142A (Fig 4A). We observed that WT and *NE⁻/⁻CatG⁻/⁻Pr3⁻/⁻* neutrophils exhibited similar DNA decondensation levels (Fig 4A), but also that *NE⁻/⁻CatG⁻/⁻Pr3⁻/⁻* murine neutrophils underwent a similar degree of pyroptosis and released similar amounts of IL-1β compared to their WT counterparts (Fig 4B). This suggested that neutrophil Serine proteases do not play a major role in inflammasome-induced neutrophil DNA delobulation and decondensation.

Next, we analyzed the importance of histone citrullination as a potential effector mechanism of inflammasome-driven DNA decondensation in neutrophils. Microscopy analysis of

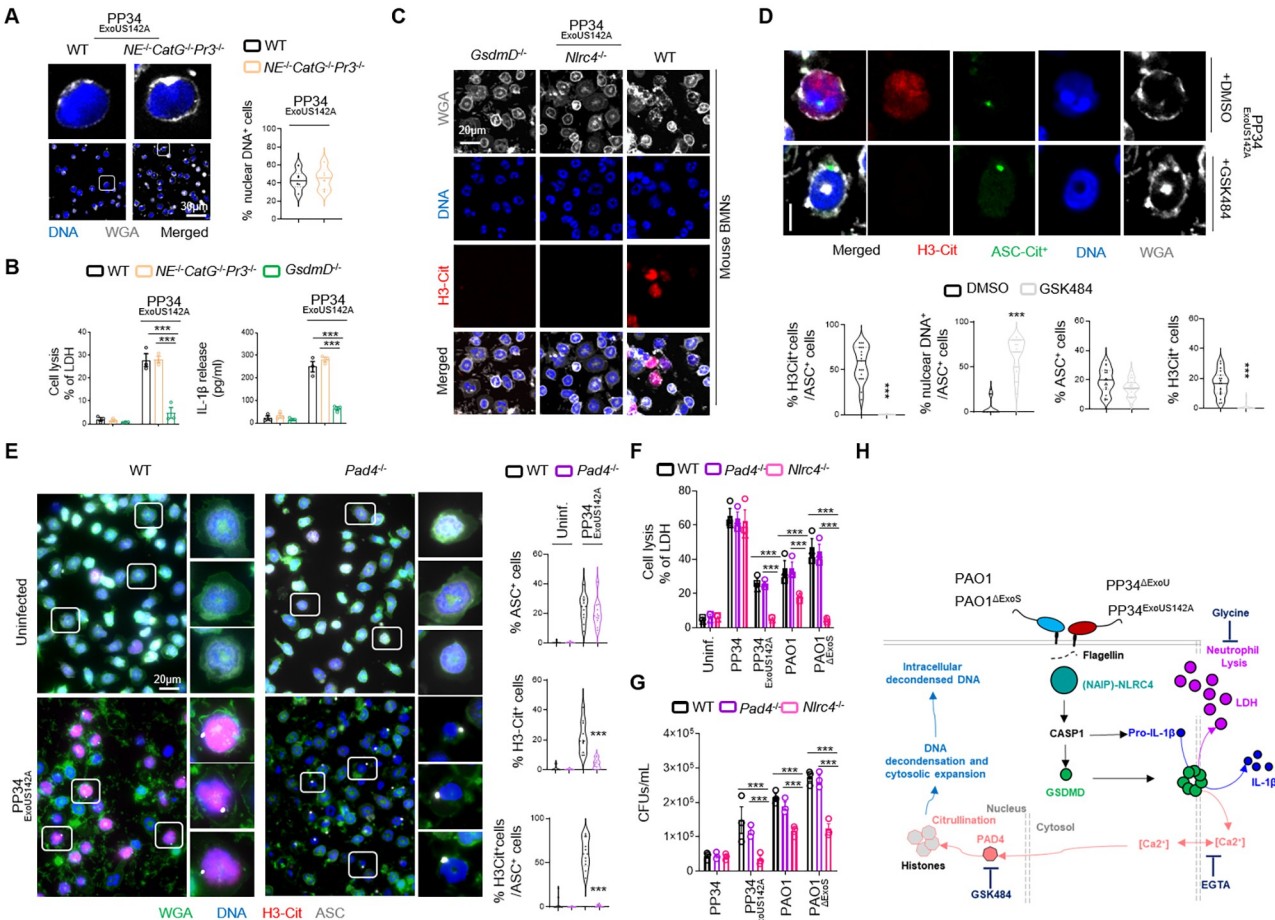

**Fig 4. Canonical inflammasome activation-induced DNA decondensation requires PAD4-dependent Histone Citrullination and occurs independently of cell shrinkage. A.** Confocal microscopy observations and quantifications of WT and $NE^{-/-}CatG^{-/-}Pr3^{-/-}$ BMNs infected for 3 hours with *P. aeruginosa* pyroptotic strain PP34$^{ExoUS142A}$ (MOI 2) and harboring decondensed DNA and plasma membrane (WGA). Nucleus (blue) was stained with Hoechst; plasma membrane is in grey (WGA staining). For quantifications, the percentage of cells with nuclear DNA was determined by quantifying the ratios of cells positives for decondensed DNA on nuclear DNA. 6 fields from n = 3 independent experiments were analyzed. Scale bar 30μm. ***p ≤ 0.001, T-test with Bonferroni correction. **B.** Measure of cell lysis (release of LDH) and IL-1β release in WT, $NE^{-/-}CatG^{-/-}Pr3^{-/-}$ and $GsdmD^{-/-}$ murine Bone Marrow Neutrophils (BMNs) infected for 3 hours with *Pseudomonas aeruginosa* pyroptotic strain PP34$^{ExoUS142A}$ (MOI 2). ***p ≤ 0.001, Two-Way Anova with multiple comparisons. Values are expressed as mean ± SEM. Graphs show combined values from three independent experiments. **C.** Confocal microscopy observations of WT, $Nlrc4^{-/-}$ and $GsdmD^{-/-}$ BMNs infected for 3 hours with *P. aeruginosa* pyroptotic strain PP34$^{ExoUS142A}$ (MOI 2) and harboring Citrullinated Histone 3 (H3-Cit), decondensed DNA (Hoechst) and plasma membrane (WGA). Nucleus (blue) was stained with Hoechst; Citrullinated Histone-3 is in red (anti H3-Cit); plasma membrane is in grey (WGA staining). Scale bar 20μm. Images are representative of one experiment performed three times with at least 150 neutrophils observed/ experiment. **D.** Confocal microscopy observations and quantifications of the percentage of cells harboring ASC complexes, H3Citrullination and nuclear/decondensed DNA in WT-ASC-Citrine$^+$ BMNs infected for 3 hours with PP34$^{ExoUS142A}$ in presence/absence of the PAD4 inhibitor GSK484 (10μM). Nucleus (blue) was stained with Hoechst; Histone-3 Citrullination is in red (Anti-H3Cit staining); plasma membrane is in grey (WGA staining). Scale bar 10μm. For quantifications, the percentage of cells with ASC complexes, nuclear DNA or positives for H3Cit (H3-Cit$^+$) was determined by quantifying the ratios of cells positives for ASC speckles, nuclear DNA or H3Cit. At least 10 fields from n = 3 independent experiments were analyzed. Values are expressed as mean ± SEM. ***p ≤ 0.001, T-test with Bonferroni correction. **E.** Confocal microscopy observations and quantifications of WT and $Pad4^{-/-}$ BMNs infected for 3 hours with *P. aeruginosa* pyroptotic strain PAO1$^{ΔExoS}$ (MOI 10) and harboring ASC complexes, Citrullinated Histone 3 (H3-Cit), decondensed DNA and plasma membrane (WGA). Nucleus (blue) was stained with Hoechst; Histone-3 Citrullination is in red (Anti-H3Cit staining); plasma membrane is in green (WGA staining), ASC is in grey (Anti-ASC). Scale bar 10μm. For quantifications, the percentage of cells with ASC complexes, nuclear DNA or positives for H3Cit (H3-Cit$^+$) was determined by quantifying the ratios of cells positives for ASC speckles, nuclear DNA or H3Cit. At least 10 fields from n = 3 independent experiments were analyzed. Values are expressed as mean ± SEM. ***p ≤ 0.001, Two-Way Anova with multiple comparisons. **F.** Measure of cell lysis (release of LDH) and bacterial killing in WT, $Pad4^{-/-}$ and $Nlrc4^{-/-}$ murine Bone Marrow Neutrophils (BMNs) infected for 3 hours with *Pseudomonas aeruginosa* strains PP34 (MOI 2), PP34$^{ExoUS142A}$ (MOI 2), PAO1 (MOI 10) or PAO1$^{ΔExoS}$ (MOI 10). ***p ≤ 0.001, Two-Way Anova with multiple comparisons. Values are expressed as mean ± SEM. LDH graph shows combined values from three independent experiments and CFU graph is from one experiment representative of three independent experiments. **G.** Overview of the different steps induced in neutrophils during NLRC4 inflammasome activation by various *Pseudomonas aeruginosa* strains.

histone citrullination in WT, *Nlrc4*⁻ᐟ⁻ and *GsdmD*⁻ᐟ⁻ showed that the fully pyroptotic strain PP34^ExoUS142A induced robust histone3-citrullination in an NLRC4- and GSDMD-dependent manner (Fig 4C). Similar experiments performed in human blood neutrophils highlighted that both pyroptotic strains PP34^ExoUS142A and PAO1^ΔExoS also promoted histone citrullination, a process that was inhibited by the use of the Caspase-1 inhibitor Z-YVAD (S4A Fig). In order to further validate these observations, we performed immunoblotting against citrullinated histones in WT, *Nlrc4*⁻ᐟ⁻ and *GsdmD*⁻ᐟ⁻ neutrophils infected with the pyroptotic *P. aeruginosa* strains PAO1, PAO1^ΔExoS, PP34^ΔExoU or with the NETotic strain PP34. As control, we used Ionomycin, a known inducer of histone citrullination-driven NETosis. We observed that Ionomycin triggered NLRC4-independent histone citrullination in neutrophils (S4B Fig). To the contrary, pyroptotic strains of *P. aeruginosa* PAO1^ΔExoS, PP34^ΔExoU and to a lower extend PAO1 all triggered NLRC4- and GSDMD-dependent histone citrullination (S4B Fig). We noticed that in *Nlrc4*⁻ᐟ⁻ and *GsdmD*⁻ᐟ⁻ neutrophils, citrullination was not fully abrogated, suggesting that minor alternative pathways might also promote histone citrullination upon infection with *P. aeruginosa* pyroptotic strains (S4B Fig).

As PAD4 is an enzyme that promotes histone citrullination and subsequent DNA decondensation in various contexts, we next tested the importance of PAD4 in inflammasome-driving DNA decondensation in neutrophils. Infection of ASC-Citrine BMNs with the pyroptotic strain PP34^ExoUS142A revealed that DNA decondensation required PAD4 as pharmacological inhibition (GSK484) of PAD4 abrogated both histone citrullination as well as nuclear DNA release (Fig 4D). In addition, measure of ASC specks (ASC⁺) in ASC-Citrine BMNs highlighted that PAD4 inhibition did not inhibit inflammasome assembly upon PP34^ExoUS142A infection (Fig 4D). Similarly, infection of WT or *Pad4*⁻ᐟ⁻ neutrophils with the pyroptotic strain PP34^ExoUS142A showed a strong defect in histone citrullination and DNA decondensation in *Pad4*⁻ᐟ⁻ neutrophils (Fig 4E). As observed previously, the induction of ASC specks was not modified in *Pad4*⁻ᐟ⁻ neutrophils, suggesting that PAD4 drives histone citrullination and DNA decondensation but not inflammasome assembly in response to PP34^ExoUS142A (Fig 4E). Further infections of WT, *Pad4*⁻ᐟ⁻ and *Nlrc4*⁻ᐟ⁻ neutrophils with pyroptotic strains PP34^ExoUS142A and PAO1^ΔExoS also showed that WT and *Pad4*⁻ᐟ⁻ neutrophils underwent a similar degree of NLRC4-dependent pyroptosis, hence excluding that PAD4 might directly modulate assembly of the NLRC4 inflammasome upon *P. aeruginosa* infection (Fig 4F).

PAD4 activation has been described to require Calcium [47]. As GSDMD pores can also promote Calcium signaling, we wondered about the importance of Calcium at regulating inflammasome-driven PAD4-dependent histone citrullination in neutrophils. We infected WT and *GsdmD*⁻ᐟ⁻ neutrophils with the pyroptotic strains PP34^ExoUS142A and PAO1^ΔExoS in presence/absence of the extracellular Calcium chelator EGTA and measured neutrophil lysis, IL-1β release as well as histone citrullination and DNA decondensation (S4C and S4D Fig). Microscopy observations showed that infection of neutrophils with the pyroptotic strain PP34^ExoUS142A efficiently induced histone citrullination and DNA decondensation in WT neutrophils, but not in *GsdmD*⁻ᐟ⁻ neutrophils (S4C Fig). This was associated with a strong defect in *GsdmD*⁻ᐟ⁻ neutrophils to undergo pyroptosis and to release IL-1β (S4D Fig). Importantly, extracellular Calcium chelation with EGTA strongly impaired both histone citrullination and DNA decondensation upon PP34^ExoUS142A infection, suggesting a strong role for extracellular Calcium in driving PAD4 activation upon inflammasome activation in neutrophils (S4C and S4D Fig). Strikingly, neutrophil pyroptosis and IL-1β release were not modified by the use of EGTA upon infection with PP34^ExoUS142A or PAO1^ΔExoS pyroptotic strains, suggesting that Calcium fluxes are important players of histone citrullination and DNA decondensation downstream of inflammasome activation in neutrophils (S4C and S4D Fig).

As inflammasome signaling in neutrophils triggers Calcium and PAD4-dependent histone citrullination and DNA decondensation, we next aimed to determine if neutrophil shrinkage, the final step of cell lysis, could regulate Calcium- and PAD4-driven DNA decondensation. To this end, we made use of the osmoprotectant Glycine [48], which is able to protect cell integrity and to inhibit LDH release without affecting inflammasome-driven IL-1β release (S4E and S4F Fig). Infection of WT and $GsdmD^{-/-}$ neutrophils with PP34$^{ExoUS142A}$ showed that Glycine did not modify neutrophil histone citrullination and DNA decondensation upon PP34$^{ExoUS142A}$ infection, although it efficiently protected cells from LDH release without affecting IL-1β release (S4E and S4F Fig). Again, $GsdmD^{-/-}$ neutrophils showed strong resistance to DNA decondensation upon PP34$^{ExoUS142A}$ infection (S4E and S4F Fig). This suggests that Calcium/PAD4-induced efficient DNA decondensation occurs in a Gasdermin-D-dependent manner, but is uncoupled from complete neutrophil lysis upon inflammasome activation.

Finally, to determine if *P. aeruginosa*-induced neutrophil pyroptosis and PAD4-dependent DNA decondensation play a microbicidal function, we infected WT, $Pad4^{-/-}$ and $Nlrc4^{-/-}$ BMNs with various *Pseudomonas* strains and evaluated their cell-autonomous immune capacities. $Nlrc4^{-/-}$ BMNs had improved ability to restrict PAO1, PP34$^{ExoUS142A}$ and PAO1$^{ΔExoS}$ infection than WT and $Pad4^{-/-}$ neutrophils (Fig 4G), suggesting that neutrophil pyroptosis more than PAD4-driven DNA decondensation promotes neutrophil failure to restrict PAO1, PP34$^{ExoUS142A}$ and PAO1$^{ΔExoS}$. Altogether, our results describe a novel pathway wherein activation of the neutrophil NLRC4 inflammasome triggers the generation of Calcium/PAD4-dependent intracellular, but not extracellular, DNA structures in a GSDMD-dependent, yet cell shrinkage-independent, manner (Fig 4H).

## Neutrophil Caspase-1 contributes *in vivo* to IL-1β production and to mouse susceptibility to *Pseudomonas aeruginosa* lung infection

Our results showed that various *P. aeruginosa* strains trigger a Caspase-1-dependent neutrophil pyroptosis *in vitro* (Figs 1–4). Based on these findings, we next sought to understand the specific role of neutrophil Caspase-1 in *P. aeruginosa* infected animals. First, to determine if neutrophils undergo Caspase-1-dependent DNA decondensation without induction of extracellular NET formation *in vivo*, we infected MRP8-GFP$^+$ (granulocytes, including neutrophils express GFP) mice and monitored for the granulocyte death features using intravital microscopy (Fig 5A). Although necrotic granulocytes exhibited NETotic features (e.g. extracellular DNA) upon exposure to the NETotic strain PP34, infection with the pyroptotic strain PP34$^{ExoUS142A}$ led to the appearance of swelled-round necrotic granulocytes that exhibited intracellular decondensed DNA, similarly to what we observed *in vitro* (Fig 5A and S5 Movie). This suggests that upon lung infection, Caspase-1-induced neutrophil pyroptosis is well occurring and displays morphological and immunological characteristics that are distinct to NETs.

Next, we sought to determine the importance of neutrophil Caspase-1 in response to *Pseudomonas aeruginosa* infection. Thus, we intranasally infected mice lacking CASP1 expression in the granulocytic compartment (MRP8$^{Cre+}$$Casp1^{flox}$) and their respective controls (MRP8$^{Cre-}$$Casp1^{flox}$) with *ExoS*- or *ExoU*-expressing *P. aeruginosa* (respectively PAO1 and PP34) or with their isogenic mutants PAO1$^{ΔExoS}$ and PP34$^{ExoUS142A}$, both of which mostly triggered a Caspase-1-dependent neutrophil pyroptosis *in vitro*. We observed that, upon lung infections with PP34, MRP8$^{Cre+}$$Casp1^{flox}$ mice did not show any differences in bacterial elimination or IL1β production, thus confirming previous work that *ExoU*-expressing *Pseudomonas aeruginosa* strains promote successful infection independently of the inflammasome pathways (Fig 5B–5D) [49, 50]. To the contrary, MRP8$^{Cre+}$$Casp1^{flox}$ mice infected with PAO1, showed a

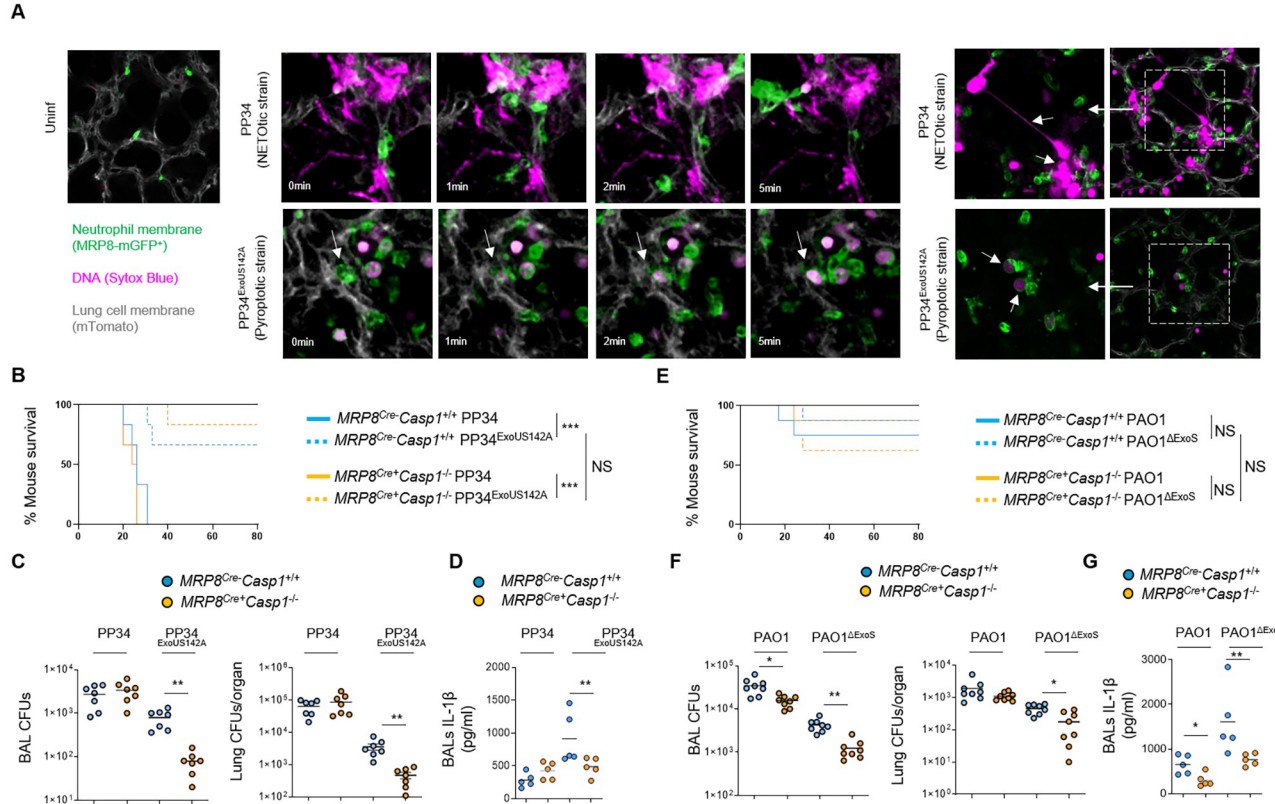

**Fig 5. Neutrophil Caspase-1 contributes to IL-1β production and to *Pseudomonas aeruginosa* spread in mice. A.** Intravital microscopy visualization of granulocyte death in MRP8-GFP+ mice infected with $2.5.10^5$ CFUs of PP34 (NETosis-inducing strain) or PP34$^{ExoUS142A}$ (pyroptotic strain) in presence of SYTOX Blue for 10 hours. Granulocyte death was observed in infected lungs by the appearance of SYTOX blue fluorescence. Pseudo colors represent vessels (gray, mTG); Granulocytes (Green, MRP8-GFP+); Dead cells (Purple, SYTOX blue). Scale bar: 20μm. Data show one experiment representative of 5 independent mice. **B.** Survival of MRP8$^{Cre-}$ *Casp1*$^{flox}$ and MRP8$^{Cre+}$*Casp1*$^{flox}$ mice intranasally infected with $5.10^5$ CFUs of PP34 (NETotic strain) or PP34$^{ExoUS142A}$ (pyroptotic strain) for 24 hours (n = 6 animals per condition). Graphs represent one experiment (6 mice/group) out of three independent *in vivo* experiments. Log-rank Cox-Mantel test was used for survival comparisons. ***p ≤ 0.001. NS; Not significant. **C.** Bronchoalveolar (BAL) and lung bacterial loads (colony forming units, CFUs) in MRP8$^{Cre-}$ *Casp1*$^{flox}$ and MRP8$^{Cre+}$*Casp1*$^{flox}$ mice intranasally infected with $2.5.10^5$ CFUs of PP34 (NETotic strain) or PP34$^{ExoUS142A}$ (pyroptotic strain) for 24 hours. Graphs represent one experiment (7 mice/group) out of three independent *in vivo* experiments; **p ≤ 0.01, Mann-Whitney analysis test. **D.** Determination of IL-1β levels in Bronchoalveolar Fluids (BALFs) MRP8$^{Cre-}$ *Casp1*$^{flox}$ and MRP8$^{Cre+}$*Casp1*$^{flox}$ mice at 10 hours after intranasal infection with $5.10^5$ CFUs (5 mice/group) of PP34 (NETotic strain) or PP34$^{ExoUS142A}$ (pyroptotic strain). Graphs represent one experiment (5 mice/group) out of three independent *in vivo* experiments; **p ≤ 0.01, Mann-Whitney analysis test. **E.** Survival of MRP8$^{Cre-}$ *Casp1*$^{flox}$ and MRP8$^{Cre+}$*Casp1*$^{flox}$ mice intranasally infected with $1.10^7$ CFUs of PAO1 (NETotic and pyroptotic strain) or PAO1$^{ΔExoS}$ (pyroptotic strain) for 24 hours (n = 6 animals per condition). Graphs represent one experiment (6 mice/group) out of three independent *in vivo* experiments. Log-rank Cox-Mantel test was used for survival comparisons. NS; Not significant. **F.** Bronchoalveolar (BAL) and lung bacterial loads (colony forming units, CFUs) in MRP8$^{Cre-}$ *Casp1*$^{flox}$ and MRP8$^{Cre+}$*Casp1*$^{flox}$ mice intranasally infected with $5.10^6$ CFUs of PAO1 (NETotic and pyroptotic strain) or PAO1$^{ΔExoS}$ (pyroptotic strain) for 24 hours. Graphs represent one experiment (7 mice/group) out of three independent *in vivo* experiments; *p ≤ 0.05, **p ≤ 0.01, Mann-Whitney analysis test. **G.** Determination of IL-1β levels in Bronchoalveolar Fluids (BALFs) MRP8$^{Cre-}$ *Casp1*$^{flox}$ and MRP8$^{Cre+}$*Casp1*$^{flox}$ mice at 8 hours after intranasal infection with $5.10^6$ CFUs (5 mice/group) of PAO1 (NETotic and pyroptotic strain) or PAO1$^{ΔExoS}$ (pyroptotic strain). Graphs represent one experiment (5 mice/group) out of three independent *in vivo* experiments; **p ≤ 0.01, Mann-Whitney analysis test.

slight but significant improved bacterial elimination in Bronchoalveolar Fluids (BALF) and lungs, a phenotype that was further amplified upon infection with PAO1$^{ΔExoS}$ and PP34$^{ExoUS142A}$ pyroptotic strains (Fig 5B–5G). Furthermore, IL-1β levels in BALFs were decreased in MRP8$^{Cre+}$*Casp1*$^{flox}$ mice infected with PAO1, PAO1$^{ΔExoS}$ and PP34$^{ExoUS142A}$. Although we cannot exclude that lower bacterial loads in MRP8$^{Cre+}$*Casp1*$^{flox}$ mice could influence the levels of IL-1β measured in BALFs, those results suggest that neutrophil Caspase-1 is also a contributor of IL-1β production upon PAO1, PAO1$^{ΔExoS}$ and PP34$^{ExoUS142A}$ infections

(Fig 5D and 5G). Altogether, our results highlight that neutrophil Caspase-1 contributes to both IL-1β release and mouse susceptibility to several *P. aeruginosa* strains.

## Discussion

Our study initially aimed at determining if neutrophil resistance to undergo Caspase-1-dependent pyroptosis can be overcome upon bacterial infection. Screening various inflammasome-activating bacteria [5, 28, 29], we found that *P. aeruginosa* successfully trigger murine and human neutrophil pyroptosis through the engagement of the fully competent canonical NLRC4 inflammasome, a process that requires T3SS-dependent injection of bacterial Flagellin. Although those results show a clear ability of neutrophils to undergo Caspase-1-dependent pyroptosis upon *P. aeruginosa* infection, our investigations performed with mice specifically lacking Caspase-1 in the granulocyte compartment do suggest a minor contribution of Caspase-1 upon *P. aeruginosa* infection. Related to this, previous studies showed a robust contribution of NLRC4 at promoting *P. aeruginosa* spread in various organs, which suggests that macrophages or other NLRC4-expressing cells are stronger contributors of mouse susceptibility to *P. aeruginosa* [51, 52]. Related to this, NLRC4-driven both IL-1β and IL-18 release upon *P. aeruginosa* infection has been linked to an altered IL-17 production and a defect of mice to efficiently control *P. aeruginosa* [51, 52]. Here, we speculate that the contribution of neutrophils at releasing IL-1β might also help the deviated immune response of the host to *P. aeruginosa*, hence leading to a better survival of this bacterium in host tissues. Another guess relies on the preferred extracellular tropism of *P. aeruginosa*. Hence, one could speculate that T3SS-injected Flagellin directly through the neutrophil plasma membrane might trigger an inefficientpyroptosis and favors *P. aeruginosa* escape from microbial trapping and/or killing [53].

Although neutrophils show intrinsic resistance to NLRC4-dependent pyroptosis (e.gs. ESCRT machinery, Caspase-1 expression levels, Ragulator pathway) [3, 54–56], the unique ability of *P. aeruginosa* strains and isogenic mutants (PAO1, CHA, PAO1$^{\Delta ExoS}$, PP34$^{\Delta ExoU}$) to trigger neutrophil pyroptosis suggests that several neutrophil factors could restrict the ability of other bacteria to promote NLRC4-dependent pyroptosis. Seminal study from Zychlynsky and colleagues found that neutrophil serine proteases could degrade the Type-3 Secretion System and flagellin virulence factors of *Shigella flexneri* [57], hence limiting their ability to hijack the neutrophil autonomous immunity and restraining *Shigella*-induced neutrophil necrosis. The lack of differences of pyroptosis induction between WT and $NE^{-/-}CatG^{-/-}Pr3^{-/-}$ neutrophils upon *P. aeruginosa* suggests that, at least in murine neutrophils, those components do not regulate *P. aeruginosa*-driven NLRC4-dependent neutrophil pyroptosis. Similarly, upon *P. aeruginosa* infection, mouse neutrophils deficient for the NADPH oxidase enzyme Nox2 undergo increased Caspase-1-dependent pyroptosis [44]. Supporting this, Warnatsch et al., could link the extracellular Oxygen Reactive Species (ROS) localization in neutrophils exposed to *Candida albicans* to an exacerbated IL-1β production and Caspase-1 activation whereas intracellular ROS had an inhibitory effect on IL1β production and Caspase-1 activity in neutrophils [58]. Whether this could explain the capacity of the extracellularly-adapted *Pseudomonas aeruginosa* to specifically promote Caspase-1-dependent neutrophil pyroptosis but not intracellular-adapted bacterial pathogens such as *Shigella* or *Salmonella* will require further investigations.

Related to this, studies showing that activation of the neutrophil NLRC4 inflammasome by *S.* Typhirmurium or the opportunistic pathogen *B. thaïlandensis* did not promote canonical pyroptosis also showed that Caspase-11-detected LPS could trigger neutrophil pyroptosis in response to those bacteria. This suggests that, in absence of efficient canonical inflammasome-driven pyroptosis, Caspase-11 might compensate at triggering neutrophil pyroptosis upon

Gram-negative bacterial infections. Here, the lack of Caspase-11-dependent pyroptosis in neutrophils upon *P. aeruginosa* infection suggests that NLRC4 inflammasome-induced neutrophil pyroptosis upon *P. aeruginosa* infection is well effective and sufficient.

Striking to us was the observation that expression of the key virulence factors ExoS or ExoU strongly influences neutrophils to go into NETosis whereas strains lacking ExoU or ExoS induced a complete rewiring of neutrophil death toward a Caspase-1-driven way. We hypothesize that the potent cytotoxic effect of these toxins towards various cell types, including neutrophils, may overcome inflammasome detection of *Pseudomonas aeruginosa* and triggers other neutrophil death programs [42, 43, 59–61]. Another non mutually exclusive guess is that these toxins directly interfere with the activation of the inflammasome pathway, thus removal of such toxins leads to an exacerbated inflammasome-response as previously reported for ExoU [40] and ExoS [62].

Although NLRC4 activation leads to Caspase-1 dependent neutrophil pyroptosis, in a pathway similar to what was previously reported in macrophage [40, 52], neutrophil pyroptosis exhibits a unique feature characterized by nuclear membrane rupture, DNA decondensation and expansion within cell cytosol. One key enzyme responsible for this morphological characteristic of neutrophil pyroptosis is the Protein arginine deiminase 4 (PAD4), activated by Calcium fluxes. Indeed, similar to the process induced by various NETosis inducers, Caspase-1 also promoted Calcium- and PAD4-dependent Histone Citrullination, which stimulated DNA relaxation and release from the nucleus but, surprisingly, not its extracellular expulsion. Why upon Caspase-11 [3], MLKL [21], NADPH [17] or NE/CatG/Pr3 [15] stimulation but not upon Caspase-1 activation neutrophils generate two different types of DNA structures remains yet to be investigated. Recently, Thiam et al., [18] observed that pharmacological stabilization of F-actin allowed the development of this "incomplete/aborted NETosis" upon Ionomycin-exposure. Interestingly, neutrophil elastase has also been shown to degrade actin [63], hence ensuring complete NETosis process. This, suggests that efficient actin degradation and/or depolymerization may be an essential player of extracellular DNA release, which would imply that the final step of NETosis might actually be a cell-regulated process involving various controllers [14, 20]. Interestingly, Chen and colleagues recently found that upon infection with *Yersinia*, murine neutrophils induce a pyroptotic program that involves virulence-inhibited innate immune sensing, hence promoting RIPK1-induced Caspase 3-dependent Gasdermin E cleavage and activation and pyroptosis [64], a process that does not trigger NETosis. This further suggests that multiple Gasdermins can trigger neutrophil pyroptosis through multiple molecular pathways and promote different morphological outcomes of neutrophils.

Regarding the immunological purpose of Caspase-1-induced neutrophil pyroptosis, we hypothesize that the decondensation of DNA but its conservation into the intracellular space might be a physical mean for neutrophils to trap some intracellular DAMPs, hence avoiding their passive release and a too strong exacerbation of the inflammatory response. Supporting this, we observed that DNA-bound Histones mostly remain trapped intracellularly, but not HMGB1 alarmin, both initially located in the nucleus. In the light of the recent discovery from Kayagaki and colleagues on the role for Ninjurin-1 at promoting active cell shrinkage and HMGB1/nucleosome DAMP release downstream of GSDMD pores in macrophages, the use of Ninjurin-1 deficient mice are full of promises [65]. Another hypothesis is that canonical inflammasome-induced neutrophil pyroptosis keeps DNA intracellularly to generate "Intracellular Traps" according to the "Pyroptosis-induced Intracellular Traps" model in macrophages described by Miao and colleagues [53]. In such a speculative model, intracellular pathogens, in addition to intracellular toxic DAMPs (e.g. Histones, DNA) might remain intracellularly, hence limiting both microbial spread and pathological DAMP-dependent inflammation.

Altogether, our results unveil the ability of neutrophils to undergo Caspase-1-dependent pyroptosis upon *Pseudomonas aeruginosa* infection which is associated with IL-1ß and soluble alarmin release in the extracellular environment, while maintaining decondensed DNA/citrullinated histone complexes intracellularly, hence expanding the spectrum of neutrophil death mechanisms. Finally, we show that this mechanism contributes to host susceptibility to *P. aeruginosa* infection *in vivo*.

## Methods

### Ethics statement

License to use human samples is under legal agreement with the EFS; contract n˚ 21PLER2017-0035AV02 after "Haute Garonne ethical committee reviewing", according to Decret N˚ 2007–1220 (articles L1243-4, R1243-61).

Mice experiments are under legal authorizations APAFIS#8521–2017041008135771 and APAFIS#12812–2018031218075551 after "IPBS and regional ethical committee evaluation", according to the local, French and European ethic laws.

All reagents, concentrations of use and their references are listed in S1 Table

### Mice

$Nlrc4^{-/-}$ [66], $Nlrp3^{-/-}$ [67], $ASC^{-/-}$, $Casp1^{-/-}$, $Casp11^{-/-}$ [68, 69], $Casp1^{-/-}Casp11^{-/-}$ [68, 69], $GsdmD^{-/-}$, $Aim2^{-/-}$, $Pad4^{-/-}$, $MRP8^{Cre+}GFP^{+}$, $MRP8^{Cre+}Casp1^{flox}$ were generated and described in previous studies. Mice were bred at the IPBS (Toulouse, France) and INRAE (Tours Nouzilly, France) animal facilities in agreement to the EU and French directives on animal welfare (Directive 2010/63/EU). Charles Rivers provided WT C57BL/6 mice.

### $MRP8^{Cre}Casp1^{flox}$ mice genotyping

$Casp1^{flox/flox}$ mice were kindly provided by Mo Lamkanfi and were crossed to $MRP8^{Cre}$ mice to generate $MRP8^{Cre}Casp1^{flox}$. *Caspase-1* genotyping was performed using Primer Fw: CGAGGGTTGGAGCTCAAGTTGACC and Primer Rv: CACTTTGACTTCTCTAAGGA CAG. *Cre* genotyping was performed using Primers Fw: CGCCGTAAATCAATCGATGA GTTGCTTC and Primers Rv: GATGCCGGTGAACGTGCAAAACAGGCTC.

### Bacterial cultures

*P. aeruginosa* strains (PAO1, CHA, PP34) and their isogenic mutants were grown overnight in Luria Broth (LB) medium at 37˚C with constant agitation.

*Legionella pneumophilia* (*L. pneumophilia*, strain Philadelphia-1) was cultivated at 37˚C in Charcoal Yeast Extract Buffered Medium according to ATCC recommendations.

*Francisella tularensis* spp *novicida* (*F. novicida*, strain U112) was cultivated overnight at 37˚C in BHI medium supplemented with 0.2% L-Cysteine.

*Staphylococcus aureus* (*S. aureus*, strain USA-300) was cultivated in BHI medium, overnight at 37˚C.

*Shigella flexnerii* (*S. flexnerii*, strain M90T) was cultivated overnight at 37˚C in Tryptic Soy Broth (TSB) according to ATCC recommendations.

*Salmonella* Typhimurium (*S.* Typhimurium, strain SL1344), *Burkholderia thailandensis* (*B. thailandensis*, strain E264), *Listeria* monocytogenes (*L. monocytogenes*, strain EGD), *Burkholderia cenocepaciae* (*B. cenocepaciae*, strain LMG 16656), *Escherichia coli* (*E. Coli*, strain K12), *Vibrio cholera* (*V. cholera*, strain El tor) were cultivated overnight at 37˚C in LB medium.

Bacteria were sub-cultured the next day by diluting overnight culture 1:25 and grew until reaching an optical density (OD) O.D.600 of 0.6–0.8.

Bacterial strains and their mutants are listed in S1 Table.

## Bacterial KO generation and complementation

The knockout vector pEXG2 was constructed and used based on the protocol described by Rietsch et al. [70] with the following modifications. Briefly, 700-bp sequences of the flanking regions of the selected gene were amplified by PCR with Q5 high fidelity polymerase (New England Biolabs). Then, the flanking regions were gel purified and inserted into pEXG2 plasmid by Gibson assembly [71]. The assembled plasmid was directly transformed into competent SM10λpir using Mix&Go competent cells (Zymo Research Corporation) and plated on selective LB plates containing 50 μg/mL kanamycin and 15 μg/mL gentamicin. The resulting clones were sequenced, and mating was allowed for 4 h with PAO1 strain at 37°C. The mated strains were selected for single cross over on plates containing 15 μg/mL gentamicin and 20 μg/mL Irgasan (removal of E.coli SM10 strains). The next day, some clones were grown in LB for 4 hours and streaked on 5% sucrose LB plates overnight at 30°C. *P. aeruginosa* clones were then checked by PCR for mutations. All primers were designed with Snapgene software (GSL Biotech LLC).

## Mice infections

Age and sex-matched animals (5–8 weeks old) per group were infected intranasally with $5.10^5$ (lethal doses) or $2.5.10^5$ CFUs of PP34/PP34$^{ExoUS142A}$/PP34$^{\Delta ExoU}$ or with $1.10^7$ CFUs (lethal doses) or $5.10^6$ CFUs of PAO1/PAO1$^{\Delta ExoS}$ strains suspended in 25μL of PBS. Animals were sacrificed at indicated times after infection and bronchoalveolar fluids (BALFs) and lungs were recovered. When specified, bacterial loads (CFU plating), cytokine levels (ELISA) were evaluated. No randomization or blinding were done.

## Intravital microscopy experiments

We relied on the previously published lung intravital microscopy method using an intercoastal thoracic window [72, 73], adapted at the IPBS CNRS-University of Toulouse TRI platform.

MRP8-mTmG mice (8–12 weeks old) were infected intratracheally with $5.10^5$ CFUs of *P. aeruginosa* PP34 or PP34$^{ExoUS142A}$ strains resuspended in 50μL of PBS and imaged 6 to 8 hours after infection. 50μL of 50μM solution of SYTOX blue (Life Technologies) was injected both intravenously (retroorbital) and intratracheally just before imaging to visualize extracellular DNA.

Next, mice were anesthetized with ketamine and xylazine and secured to a microscope stage. A small tracheal cannula was inserted, sutured and attached to a MiniVent mouse ventilator (Harvard Apparatus). Mice were ventilated with a tidal volume of 10 μl of compressed air (21% $O_2$) per gram of mouse weight, a respiratory rate of 130–140 breaths per minute, and a positive-end expiratory pressure of 2–3 cm $H_2O$. Isoflurane was continuously delivered to maintain anesthesia and 300 μl of 0.9% saline solution were i.p. administered in mice every hour for hydration. Mice were placed in the right lateral decubitus position and a small surgical incision was made to expose the rib cage. A second incision was then made into the intercostal space between ribs 4 and 5, through the parietal pleura, to expose the surface of the left lung lobe. A flanged thoracic window with an 8 mm coverslip was inserted between the ribs and secured to the stage using a set of optical posts and a 90° angle post clamp (Thor Labs). Suction was applied to gently immobilize the lung (Dexter Medical). Mice were placed in 30°C heated box during microscopy acquisition to maintain the body temperature and the 2-photon

microscope objective was lowered over the thoracic window. Intravital imaging was performed using a Zeiss 7MP upright multi-photon microscope equipped with a 20×/1.0 objective and a Ti-Sapphire femtosecond laser, Chameleon-Ultra II (Coherent Inc.) tuned to 920 nm. SYTOX Blue, GFP and Tomato emission signals were detected thanks to the respective bandpass filters: Blue (SP485), Green (500–550) and Red (565–610). Images were analyzed using Imaris software (Bitplane) and Zen (Zeiss).

## Isolation of primary murine neutrophils

Murine Bone marrow cells were isolated from tibias and femurs, and neutrophils were purified by positive selection using Anti-Ly-6G MicroBead Kit (Miltenyi Biotech) according to manufacturer's instructions. This process routinely yielded >95% of neutrophil population as assessed by flow cytometry of Ly6G$^+$/CD11b$^+$ cells.

## Isolation of primary human neutrophils

Whole blood was collected from healthy donors by the "Ecole française du sang" (EFS, Toulouse Purpan, France) in accordance with relevant guidelines. Written, informed consent was obtained from each donor. Neutrophils were then isolated by negative selection using MACSxpress Whole Blood Human Neutrophil Isolation Kit (Miltenyi Biotech) according to manufacturer's instructions. Following isolations cells were centrifuged 10 min at 300 g and red blood cells were eliminated using Red blood cells (RBC) Lysis Buffer (BioLegend). This procedure gives >95% of neutrophil population as assessed by flow cytometry of CD15+/CD16+ cells.

## Cell plating and treatment of Neutrophils

Following isolation, Neutrophils were centrifugated for 10 min at 300 g and pellet was resuspendent in serum free OPTI-MEM medium. Absolute cell number was determined with automated cell counter Olympus R1 with trypan blue cell death exclusion method (typically living cells represent >70% of cell solution) and cell density was adjusted at $10^6$ / mL by adding OPTI-MEM culture medium. Neutrophils were then plated in either 96 well plates, 24 well plates or 6 well plates with 100 μL ($10^5$ cells), 500 μL ($5.10^5$ cells) or 2 mL ($2.10^6$ cells) respectively. When indicated cells were incubated with chemical inhibitors Z-VAD-fmk (20 μM), Y-VAD-fmk (20μM, 40 μM), Z-DEVD-fmk (40μM), Z-IETD-fmk (40μM), Z-LEVD (40μM), GSK484 (10 μM), as indicated in each experimental setting. Neutrophils were infected with various bacterial strains and multiplicity of infections (M.O.I.) as indicated.

## Kinetic analysis of Neutrophil's permeability by SYTOX Green incorporation assay

Cells are plated at density of 1 x $10^5$ per well in Black/Clear 96-well Plates in OPTI-MEM culture medium supplemented with SYTOX-Green dye (100ng/mL) and infected/treated as mentioned in figure legend. Green fluorescence are measured in real-time using Clariostar plate reader equipped with a 37°C cell incubator. Maximal cell death was determined with whole cell lysates from unstimulated cells incubated with 1% Triton X-100.

## ELISA and plasma membrane lysis tests

Cell death was measured by quantification of the lactate dehydrogenase (LDH) release into the cell supernatant using LDH Cytotoxicity Detection Kit (Takara). Briefly, 100 μL cell supernatant were incubated with 100 μL LDH substrate and incubated for 15 min. The enzymatic reaction was stopped by adding 50 μL of stop solution. Maximal cell death was determined with

whole cell lysates from unstimulated cells incubated with 1% Triton X-100. Human and mouse IL-1β secretion was quantified by ELISA kits (Thermo Fisher Scientific) according to the manufacturer's instructions.

## Preparation of neutrophil lysates and supernatant for immunoblot

At the end of the treatment 5 mM of diisopropylfluorophosphate (DFP) cell permeable serine protease inhibitor was added to cell culture medium. Cell' Supernatant was collected and clarified from non-adherent cells by centrifugation for 10 min at 300 g. Cell pellet and adherent cells were lysed in 100 μL of RIPA buffer (150 mM NaCl, 50 mM Tris-HCl, 1% Triton X-100, 0.5% Na-deoxycholate) supplemented with 5 mM diisopropylfluorophosphate (DFP) in addition to the protease inhibitor cocktail (Roche). Cell scrapper was used to ensure optimal recovery of cell lysate. Collected cell lysate was homogenized by pipetting up and down ten times and supplemented with laemli buffer (1X final) before boiling sample for 10 min at 95˚C. Soluble proteins from cell supernatant fraction were precipitated as described previously [74]. Precipitated pellet was then resuspended in 100 μL of RIPA buffer plus laemli supplemented with 5 mM diisopropylfluorophosphate (DFP) and protease inhibitor cocktail (Roche) and heat denaturated for 10 min at 95˚C. Cell lysate and cell supernatant fraction were then analysed by immunoblot either individually or in pooled sample of lysate plus supernatant (equal vol/vol).

## Treatment of Neutrophils for Immunofluorescences

$5.10^5$ Cells were plated on 1.5 glass coverslips in 24 well plate and infected/treated as described above. At the end of the assay, cell supernatant was removed and cells were fixed with a 4% PFA solution for 10 min at 37˚C. PFA was then removed and cells were washed 3 times with HBSS. When desired, plasma membrane was stained with Wheat Germ Agglutinin, Alexa Fluor 633 Conjugate (ThermoFisher Scientifique) at 1/100th dilution in HBSS, and incubated for 30 min under 100 rpm orbital shaking conditions. Then cells were washed with HBSS and processed for further staining. Permeabilization was performed by incubating cells for 10 min in PBS containing 0.1% Triton X-100. To block unspecific binding of the antibodies cells are incubated in PBS-T (PBS+ 0.1% Tween 20), containing 2% BSA, 22.52 mg/mL glycine in for 30 min. 3 washes with PBS-T was performed between each steps. Primary antibodies staining was performed overnight at 4˚C in BSA 2%—Tween 0.1%—PBS (PBS-T) solution. Coverslips were washed three times with PBS-T and incubated with the appropriate fluor-coupled secondary antibodies for 1 hour at room temperature. DNA was counterstained with Hoechst. Cells were then washed three times with PBS and mounted on glass slides using Vectashield (Vectalabs). Coverslips were imaged using confocal Zeiss LSM 710 (INFINITY, Toulouse) or Olympus Spinning disk (Image core Facility, IPBS, Toulouse) using a 40x or a 63x oil objective. Unless specified, for each experiment, 5–10 fields (∼50–250 cells) were manually counted using Image J software.

## Scanning electron microscopy experiments

For scanning electron microscopy observations, cells were fixed with 2.5% glutaraldehyde in 0.2M cacodylate buffer (pH 7.4). Preparations were then washed three times for 5min in 0.2M cacodylate buffer (pH 7.4) and washed with distilled water. Samples were dehydrated through a graded series (25 to 100%) of ethanol, transferred in acetone and subjected to critical point drying with CO2 in a Leica EM CPD300. Dried specimens were sputter-coated with 3 nm platinum with a Leica EM MED020 evaporator and were examined and photographed with a FEI Quanta FEG250.

## ImageStreamX

Cells isolated from Bronchoalveolar (BAL) washes were pelleted by centrifugation (10 min at 300 g). Neutrophils were stained prior to fixation with anti-Ly6G (APC-Vio770, Miltenyi-Biotec Clone: REA526 | Dilution: 1:50) in MACS buffer (PBS-BSA 0,5%-EDTA 2mM) in presence of FC block (1/100) and Hoechst (1 μM). Then, cells were fixed in 4% PFA. Data were acquired on ImageStreamXMKII (Amnis) device (CPTP Imaging and Cytometry core facility) and analyzed using IDEAS software v2.6 (Amnis). The gating strategy used to evaluate inflammasome activation in neutrophils was performed as follow: (i) a gate was set on cells in focus [Cells in Focus] and (ii) a sub-gate was created on single cells [Single Cells]. Then we gated first on (iii) LY6G+ Neutrophils [LY6G+] and second on (iv) ASC-citrine+ and Hoechst+ cells [Hoechst +/ASC-Citrine+] within LY6G+ population. (v) To distinguish cells with active (ASC-speck) versus inactive inflammasome (Diffuse ASC), we plotted the Intensity with the area of ASC-citrine. This strategy allow to distinguish cells with active inflammasome that were visualized and quantified (S2D Fig).

## Statistical tests used

Statistical analysis was performed with Prism 8.0a (GraphPad Software, Inc). Otherwise specified, data are reported as mean with SEM. T-test with Bonferroni correction was chosen for comparison of two groups. Two-way Anova with multiple comparisons test was used for comparison of more than two groups. For *in vivo* mice experiments and comparisons we used Mann-Whitney tests and mouse survival analysis were performed using log-rank Cox-Mantel test. P values are shown in figures with the following meaning; NS non-significant and Significance is specified as $^*p \leq 0.05$; $^{**}p \leq 0.01$, $^{***}p \leq 0.001$.

## Supporting information

**S1 Table. List and reference of Reagents and Tools available.**
(DOCX)

**S1 Movie. Time Lapse Fluorescence microscopy of ASC-Citrine murine BMNs infected with PP34 (NETotic strain, MOI 2) for 9 hours (540 minutes).** Nucleus (blue) was stained with Hoechst; ASC is in yellow (ASC-Citrine); plasma membrane is in green (WGA staining); plasma membrane permeabilization is stained in red (cell impermanent DNA dye Propidium Iodide, PI).
(AVI)

**S2 Movie. Time Lapse Fluorescence microscopy of ASC-Citrine murine BMNs infected with PP34$^{S142}$ (Pyroptotic strain, MOI 2) for 9 hours (540 minutes).** Nucleus (blue) was stained with Hoechst; ASC is in yellow (ASC-Citrine); plasma membrane is in green (WGA staining); plasma membrane permeabilization is stained in red (cell impermanent DNA dye Propidium Iodide, PI).
(AVI)

**S3 Movie. 3D reconstruction and projection of ASC-Citrine murine BMNs infected with *P. aeruginosa* NETotic strain PP34 (MOI 2) for 3 hours.** Nucleus (blue) was stained with Hoechst; ASC is in red (ASC-Citrine); plasma membrane is in green (WGA staining).
(MP4)

**S4 Movie. 3D reconstruction and projection of ASC-Citrine murine BMNs infected with *P. aeruginosa* pyroptotic strain PP34 $^{ExoUS142}$ (MOI 2) for 3 hours.** Nucleus (blue) was stained

with Hoechst; ASC is in red (ASC-Citrine); plasma membrane is in green (WGA staining).
(MP4)

**S5 Movie. Intravital microscopy visualization of granulocyte death in MRP8-GFP$^+$ mice infected with 2.5.10$^5$ CFUs of PP34 (NETosis-inducing strains) or PP34$^{\text{ExoUS142A}}$ (pyroptosis-inducing strain) in presence of SYTOX Blue for 10 hours.** Granulocyte death was observed in infected lungs by the appearance of SYTOX blue fluorescence. Pseudo colors represent vessels (gray, mTG); Granulocytes (Green, MRP8-GFP+); Dead cells (Purple, SYTOX blue). Scale bar: 20μm.
(MP4)

**S1 Fig. Multiple *P. aeruginosa* strains trigger Caspase-1-dependent pyroptosis in neutrophils. A.** Measure of basal cell lysis (release of LDH) in WT and *Casp1$^{-/-}$* murine Bone Marrow Neutrophils (BMNs) in culture for the indicated times. Values are expressed as mean ± SEM. Graphs show combined values from three independent experiments. **B.** Measure of cell lysis (release of LDH) and IL-1β release in WT or *Casp1$^{-/-}$* murine Bone Marrow Neutrophils (BMNs) infected for 3 hours with *Pseudomonas aeruginosa* PAO1 and its isogenic mutants lacking T3SS expression (PAO1$^{\Delta\text{ExsA}}$), Flagellin (PAO1$^{\Delta\text{FliC}}$) or T3SS-derived toxins ExoS, ExoY, ExoT (PAO1$^{\Delta\text{ExoS}}$, PAO1$^{\Delta\text{ExoY}}$, PAO1$^{\Delta\text{ExoT}}$) at a multiplicity of infection (MOI) of 10. $^{***}$p ≤ 0.001, Two-Way Anova with multiple comparisons. Values are expressed as mean ± SEM. Graphs show combined values from three independent experiments. Values are expressed as mean ± SEM. **C.** Measure of bacterial uptake (Colony-forming Units, CFUs) in WT or *Casp1$^{-/-}$* BMNs infected for 45 minutes with *Pseudomonas aeruginosa* PAO1 and its isogenic mutants lacking Flagellin (PAO1$^{\Delta\text{FliC}}$), Flagellin motors MotABCD (PAO1$^{\Delta\text{MotABCD}}$) or both Flagellin and Flagellin motors MotABCD (PAO1$^{\Delta\text{FliC}/\Delta\text{MotABCD}}$) at a MOI of 10. Here, due to their lack of motility, bacteria were centrifuged for 5 min/1000 rpm to ensure neutrophil/bacterial contact. $^{***}$p ≤ 0.001, Two-Way Anova with multiple comparisons. Values are expressed as mean ± SEM. Graph show one experiment representative of three independent experiments. **D.** Measure of cell lysis (release of LDH) and IL-1β release in WT and *Casp1$^{-/-}$* murine Bone Marrow Neutrophils (BMNs) infected for the indicated times with PAO1$^{\Delta\text{MotABCD}}$ or PAO1$^{\Delta\text{FliC}/\Delta\text{MotABCD}}$ at an MOI of 10. Here, due to their lack of motility, bacteria were centrifuged for 5 min/1000 rpm to ensure neutrophil/bacterial contact. $^{***}$p ≤ 0.001, Two-Way Anova with multiple comparisons. Values are expressed as mean ± SEM. Graphs show combined values from three independent experiments. **E.** Measure of the percentage of cells with plasma membrane permeabilization over time using SYTOX Green incorporation in murine WT and *Casp1$^{-/-}$* BMNs infected with *Pseudomonas aeruginosa* mutant invalidated for ExoU catalytic activity PP34$^{\text{ExoUS142A}}$ at a multiplicity of infection (MOI) of 2. $^{***}$p ≤ 0.001, Two-Way Anova with multiple comparisons. Values are expressed as mean ± SEM. Graphs show combined values from three independent experiments. **F.** Measure of the percentage of cells with plasma membrane permeabilization over time using SYTOX Green incorporation in human blood neutrophils BMNs infected with *Pseudomonas aeruginosa* mutant invalidated for ExoU catalytic activity PP34$^{\text{ExoUS142A}}$ at a multiplicity of infection (MOI) of 2 in presence/absence of various Caspase inhibitors, Z-YVAD (Casp1 inhibitor, 20μM), Z-DEVD (Casp3/7 inhibitor, 20μM), Z-IETD (Casp8 inhibitor, 20μM). $^{***}$p ≤ 0.001, Two-Way Anova with multiple comparisons. Values are expressed as mean ± SEM. Graphs show combined values from three independent experiments.
(TIF)

**S2 Fig. *P. aeruginosa* triggers NLRC4-dependent pyroptosis in neutrophils. A.** Measure of cell lysis (release of LDH) in WT, *Casp1$^{-/-}$*, *Casp1$^{-/-}$*, *Casp11$^{-/-}$*, *Casp1$^{-/-}$Casp11$^{-/-}$*, *Nlrp3$^{-/-}$*,

$AIM2^{-/-}$, $Nlrc4^{-/-}$ and $ASC^{-/-}$ murine Bone Marrow Neutrophils (BMNs) infected for 3 hours with *Pseudomonas aeruginosa* pyroptotic strains PP34$^{ExoUS142A}$ (MOI 2) and PAO1$^{\Delta ExoS}$ (MOI 10). ***p ≤ 0.001, Two-Way Anova with multiple comparisons. Values are expressed as mean ± SEM. Graphs show combined values from three independent experiments. **B.** Gating strategy used to evaluate inflammasome activation in neutrophils was performed as follow: (i) a gate was set on cells in focus [Cells in Focus] and (ii) a sub-gate was created on single cells [Single Cells]. Then we gated first on (iii) LY6G$^+$ Neutrophils [LY6G$^+$] and second on (iv) ASC-citrine$^+$ and Hoechst$^+$ cells [Hoechst$^+$/ASC-Citrine$^+$] within LY6G$^+$ population. (v) To distinguish cells with active (ASC-speck) versus inactive inflammasome (Diffuse ASC), we plotted the Intensity with the area of ASC-citrine. This strategy allow to distinguish cells with active inflammasome that were visualized and quantified.
(TIF)

**S3 Fig. NLRC4-dependent neutrophil pyroptosis associates to DNA decondensation. A.** Confocal microscopy observations and quantification of primary human blood neutrophils infected for 3 hours with *P. aeruginosa* pyroptotic strain PP34$^{ExoUS142A}$ (MOI 2) and harboring ASC complexes, decondensed DNA and plasma membrane. Nucleus (blue) was stained with Hoechst; ASC is in Red (anti-ASC); Plasma membrane is in grey (WGA). Scale bar 10μm. Images are representative of one experiment performed three times with at least 6–10 fields neutrophils observed/ quantified for ASC specks ratios. ***p ≤ 0.001, T-test with Bonferroni correction. **B.** Immunoblotting observation of Histone 3, HMGB1, Lamin B1, GAPDH, Actin, Gasdermin D (GSDMD) and NLRC4 in cellular soluble and insoluble fractions as well as in the supernatant from WT and $Nlrc4^{-/-}$ murine BMNs infected with *P. aeruginosa* NETotic strain PP34 or pyroptotic strain PP34$^{ExoUS142A}$ (MOI 2) for 3 hours. Immunoblots show one experiment performed two times. **C.** Overview of the different steps induced in neutrophils during NLRC4 inflammasome activation by various *Pseudomonas aeruginosa* strains.
(TIF)

**S4 Fig. Inflammasome signalling induce neutrophil DNA decondensation through Calcium-dependent, but cell lysis-independent, manner. A.** Confocal microscopy observations and quantifications of human blood neutrophils infected for 3 hours with PAO1$^{\Delta ExoS}$ (MOI 10) or PP34$^{ExoUS142A}$ (MOI 2) in presence/absence of Caspase-1 inhibitor Z-YVAD (20μM) and harboring Citrullinated Histone 3 (H3-Cit), decondensed DNA and nuclear membrane (LaminB1). Nucleus (blue) was stained with Hoechst; Citrullinated Histone-3 is in red (anti H3-Cit); plasma membrane is in grey (WGA staining). Scale bar 10μm. Images are representative of one experiment performed three times with at least 150 neutrophils observed/ experiment. **B.** Immunoblotting of Citrullinated Histone 3 (H3Cit), total Histone 3 and preformed and cleaved Gasdermin-D (p55/p30) in WT, $Nlrc4^{-/-}$ and $GsdmD^{-/-}$ BMNs treated with Ionomycin (10μM, 3 hours) or infected for 3 hours with PAO1, PAO1$^{\Delta ExoS}$, PAO1$^{ExsA-}$ (MOI 10) or with PP34, PP34$^{\Delta ExoU}$ (MOI 2). Immunoblots show combined lysates and supernatants from one experiment performed three times. **C.** Confocal microscopy observations and quantifications of WT and $GsdmD^{-/-}$ BMNs infected for 3 hours with *P. aeruginosa* pyroptotic strain PAO1$^{ExoUS142A}$ (MOI 2) in presence/absence of EGTA (10mM) and harboring Citrullinated Histone 3 (H3-Cit) and decondensed DNA. Nucleus (blue) was stained with Hoechst; Histone-3 Citrullination is in red (Anti-H3Cit staining). For quantifications, the percentage of cells positives for H3Cit (H3-Cit$^+$) and decondensed DNA was determined by quantifying the ratios of cells positives for H3Cit and decondensed DNA over the total amount of cells. At least 6 fields from n = 3 independent experiments were analyzed. Values are expressed as mean ± SEM. ***p ≤ 0.001, Two-Way Anova with multiple comparisons. **D.** Measure of cell lysis (release of LDH) and IL-1β release in WT or $GsdmD^{-/-}$ BMNs infected for 3 hours with

PP34$^{ExoUS142A}$ (MOI2) or PAO1$^{\Delta ExoS}$ (MOI 10) in presence/absence of EGTA (10mM).
***p ≤ 0.001, Two-Way Anova with multiple comparisons. Values are expressed as
mean ± SEM. Graphs show combined values from three independent experiments. **E.** Confo-
cal microscopy observations and quantifications of WT and *GsdmD*$^{-/-}$ BMNs infected for 3
hours with *P. aeruginosa* pyroptotic strain PAO1$^{ExoUS142A}$ (MOI 2) in presence/absence of
Glycine (5mM) and harboring Citrullinated Histone 3 (H3-Cit) and decondensed DNA.
Nucleus (blue) was stained with Hoechst; Histone-3 Citrullination is in red (Anti-H3Cit stain-
ing). For quantifications, the percentage of cells positives for H3Cit (H3-Cit$^+$) and decon-
densed DNA was determined by quantifying the ratios of cells positives for H3Cit and
decondensed DNA over the total amount of cells. At least 6 fields from n = 3 independent
experiments were analyzed. Values are expressed as mean ± SEM. ***p ≤ 0.001, Two-Way
Anova with multiple comparisons. **F.** Measure of cell lysis (release of LDH) and IL-1β release
in WT or *GsdmD*$^{-/-}$ BMNs infected for 3 hours with PP34$^{ExoUS142A}$ (MOI2) or PAO1$^{\Delta ExoS}$
(MOI 10) in presence/absence of Glycine (5mM). ***p ≤ 0.001, Two-Way Anova with multi-
ple comparisons. Values are expressed as mean ± SEM. Graphs show combined values from
three independent experiments.
(TIF)

## Acknowledgments

*Nlrc4*$^{-/-}$ mice were provided by Clare E. Bryant [66] and generated by Millenium Pharmaceuti-
cal, *GsdmD*$^{-/-}$ mice [75] came from P. Broz (Univ of Lausanne, Switzerland), *Casp11*$^{-/-}$ and
*Casp1*$^{-/-}$/ *Casp11*$^{-/-}$ came from B. Py (ENS Lyon, France) and Junying Yuan (Harvard Med
School, Boston, USA) [68, 69]. Virginie Petrilli (ENS Lyon, France) provided *Nlrp3*$^{-/-}$ mice
that were generated by Fabio Martinon [67]. Thomas Henry (CIRI, Lyon, France) provided
*ASC*$^{-/-}$ and *AIM2*$^{-/-}$ mice upon agreement with Genentech (Roche, San Francisco, USA) and.
ASC-Citrine (#030744) and *Pad4*$^{-/-}$ (#030315) mice came from Jaxson Laboratory (USA) and
were generated by Douglas T Golenbock (University of Massachusetts Medical School, USA)
and Kerri Mowen (The Scripps Research Institute, USA) respectively. MRP8$^{Cre}$/*Casp1*$^{flox}$ mice
are provided by Natalie Winter (INRAE Tours Nouzilly, France) and were generated by cross-
ing MRP8$^{Cre}$ (Jackson # 021614) mice with *Caspase1*$^{flox}$ mice generated by Mohamed Lam-
kanfi (Univ. of Ghent, Belgium)[76]. MRP8$^{CreGFP}$ and mTmG mice were obtained from
Jackson laboratories and generated respectively by Emmanuelle Passegue (UCSF, USA) and
Liqun Luo, (Stanford University, USA). *Pseudomonas aeruginosa* strains were a kind gift of
Ina Attrée (CNRS, Grenoble, France). Authors also acknowledge the animal facility and
Cytometry/microscopy platforms of the INFINITY, CBI and IPBS institutes and particularly
Valerie Duplan-Eche for Imagestream acquisition and analysis.

## Author Contributions

**Conceptualization:** David Pericat, Emma Lefrançais, Etienne Meunier, Rémi Planès.

**Data curation:** David Pericat, Renaud Poincloux, Etienne Meunier, Rémi Planès.

**Formal analysis:** David Pericat, Jean-Philippe Girard, Céline Cougoule, Renaud Poincloux,
Etienne Meunier, Rémi Planès.

**Funding acquisition:** Christine T. N. Pham, Céline Cougoule, Emma Lefrançais, Etienne
Meunier, Rémi Planès.

**Investigation:** Karin Santoni, David Pericat, Leana Gorse, Julien Buyck, Miriam Pinilla, Laure
Prouvensier, Salimata Bagayoko, Audrey Hessel, Stephen Adonai Leon-Icaza, Elisabeth

Bellard, Serge Mazères, Emilie Doz-Deblauwe, Nathalie Winter, Jean-Philippe Girard, Renaud Poincloux, Emma Lefrançais, Etienne Meunier, Rémi Planès.

**Methodology:** Karin Santoni, David Pericat, Leana Gorse, Julien Buyck, Miriam Pinilla, Laure Prouvensier, Salimata Bagayoko, Audrey Hessel, Stephen Adonai Leon-Icaza, Elisabeth Bellard, Serge Mazères, Emilie Doz-Deblauwe, Christophe Paget, Jean-Philippe Girard, Renaud Poincloux, Emma Lefrançais, Etienne Meunier, Rémi Planès.

**Project administration:** Céline Cougoule, Mohamed Lamkanfi, Emma Lefrançais, Etienne Meunier, Rémi Planès.

**Resources:** Nathalie Winter, Christine T. N. Pham, Renaud Poincloux, Mohamed Lamkanfi, Emma Lefrançais, Etienne Meunier, Rémi Planès.

**Supervision:** Nathalie Winter, Christophe Paget, Renaud Poincloux, Emma Lefrançais, Etienne Meunier, Rémi Planès.

**Validation:** Karin Santoni, David Pericat, Leana Gorse, Julien Buyck, Christophe Paget, Renaud Poincloux, Mohamed Lamkanfi, Emma Lefrançais, Etienne Meunier, Rémi Planès.

**Visualization:** Leana Gorse, Julien Buyck, Renaud Poincloux, Emma Lefrançais, Rémi Planès.

**Writing – original draft:** Karin Santoni, Mohamed Lamkanfi, Emma Lefrançais, Etienne Meunier, Rémi Planès.

**Writing – review & editing:** Karin Santoni, Nathalie Winter, Christophe Paget, Christine T. N. Pham, Céline Cougoule, Renaud Poincloux, Mohamed Lamkanfi, Emma Lefrançais, Etienne Meunier, Rémi Planès.

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
