## [Decision Letter · Decision Letter 0]

11 Mar 2022

Dear Dr. Meunier,

Thank you very much for submitting your manuscript "Caspase-1-driven neutrophil pyroptosis promotes an incomplete NETosis upon Pseudomonas aeruginosa infection" for consideration at PLOS Pathogens. As with all papers reviewed by the journal, your manuscript was reviewed by members of the editorial board and by several independent reviewers. In light of the reviews (below this email), we would like to invite the resubmission of a significantly-revised version that takes into account the reviewers' comments.

The reviewers have identified experimental important issues that should be addressed. In addition, the text as written have issues with clarity. Examples include:

1. Description of results (example is Fig. 1B and Fig 2A-E). So brief that it doesn't seem to help the readers understand the data presented. Expanding these and other sections would be helpful.

2. Reference to Fig. 3E on line 265 seems incorrect.

3. Legends for some figure panels and figures can be improved for clarity. Fig 1D - are these showing merged micrographs or separate brightfield and SYTOX green images. It is not clear.

4. Overall, the authors are encouraged to have the language reviewed for clarity. A reduction in the use of quotation marks could be helpful.

We cannot make any decision about publication until we have seen the revised manuscript and your response to the reviewers' comments. Your revised manuscript is also likely to be sent to reviewers for further evaluation.

Sincerely,

Vincent T Lee

Associate Editor

PLOS Pathogens

David Skurnik

Section Editor

PLOS Pathogens

Kasturi Haldar

Editor-in-Chief

PLOS Pathogens

orcid.org/0000-0001-5065-158X

Michael Malim

Editor-in-Chief

PLOS Pathogens

orcid.org/0000-0002-7699-2064

The reviewers have identified experimental important issues that should be addressed. In addition, the text as written have issues with clarity. Examples include:

1. Description of results (example is Fig. 1B and Fig 2A-E). So brief that it doesn't seem to help the readers understand the data presented. Expanding these and other sections would be helpful.

2. Reference to Fig. 3E on line 265 seems incorrect.

3. Legends for some figure panels and figures can be improved for clarity. Fig 1D - are these showing merged micrographs or separate brightfield and SYTOX green images. It is not clear.

4. Overall, the authors are encouraged to have the language reviewed for clarity. A reduction in the use of quotation marks could be helpful.

Reviewer's Responses to Questions

**Part I - Summary**

Reviewer #1: In the present manuscript, Santoni et al describe a phenomenon of incomplete netosis triggered by inflammasome activation by certain Pseudomonas strains in neutrophils. This is interesting as neutrophils have been believed to be largely refractory to pyroptosis upon bacterial infection under normal circumstances. The authors attribute some of the observed accompanying phenomena such as histone citrunillation to PAD4 activation but fail to uncover the reason for P. aeruginosa specific induction of neutrophil pyroptosis over other bacteria. They demonstrate a role for Caspase 1 in neutrophils with a neutrophil specific Casp1-Cre mouse line in the infectivity levels of Pseudomonas in vivo, extending on previously published data with whole Caspase-1 knockouts. Their observations suggest that NETosis is a very controlled process with several unknown aspects remaining.

Several of the observations require additional quantifications to allow the conclusions drawn, and the manuscript should be carefully edited for grammar and correct referencing of figures and panels.

Reviewer #2: Inflammasomes are multiprotein complexes that activate caspase-1 and -11. In most cases, activation of these inflammatory caspases promote IL-1b maturation and a lytic form of cell death known as pyroptosis. However, in neutrophils, caspase-1 activation appears to resist pyroptosis through a poorly understood mechanism. Here, Santoni and co-workers show that neutrophils are not universally resistant to caspase-1-dependent pyroptosis as previously reported. They found that various strains of Pseudomonas aeruginosa can in fact trigger caspase-1-dependent pyroptosis in neutrophils, and some of these strains promote nuclear membrane rupture and histone citrullination in the absence of cell lysis, which the authors termed ‘incomplete NETosis’. Overall, this is an interesting manuscript and will be of interest to the microbiology and innate immunity community. However, this is a complicated study, and the presentation of the manuscript can be improved significantly. The labelling and definition require a lot of work – while the introduction, abstract and summary are quite well written, the results are very hard to follow, due to typo mistakes and inconsistent labelling. For example (also mentioned in point 5, 9), does PP34 normally express ExoU? If so, it is redundant to write PP34ExoU, since that implies exoU is introduced into the strain. Are the authors referring to the same strain when they mention ExoU- and ExoUS142A? Note that these are not the same. I suggest the to use the standard nomenclature for describing bacterial strains - (delta Greek sign)ExoU for knockout and not ExoU-, and clearly differentiate when using a catalytic inactive versus a knockout. There are also other experimental controls and concerns that should be addressed.

**Part II – Major Issues: Key Experiments Required for Acceptance**

Reviewer #1: - For the Pseudomonas strain comparison experiments shown in figure 1C and E and S1, it is important to verify that the strains are equally infectious to be able to draw any conclusion on the role of effectors. Also, aflagellated bacteria are expected to be less infectious due to their limited motility. It is not clear from the methods whether the authors i.e. spun Fla-negative bacteria onto cells to help with infection. Ideally, the authors would plate for numbers of intracellular bacteria at an early, pre-death timepoint or something similar

-Imaging in 3A, 3B and 3C needs to be quantified over multiple images. It is also not clear why the magnification for uninfected cells in 3A is different than for infected cells

-While GSK484 seems to be a very specific inhibitor, it would be nice to see the H3 citrullin staining and quantification as in 3E in Pad4 knockouts as used in the supplemental data.

Reviewer #2: Major points

1. Neutrophils are extremely short-lived and sensitive cells and should be undergoing 20-30% basal cell death at the end of inflammasome assays. Why are the values for LDH release for untreated neutrophils 0? I believe that is not possible. Authors should also indicate untreated values for all other experiments to determine the baseline of neutrophil cell death.

2. Early studies from different labs have shown that bacterial infection triggers at best 30-40% LDH release in neutrophils, a finding that the authors reproduced in Figure S2C too. Why are the values so high in Figure 1? Can the authors explain the discrepancy with the field? Are the neutrophils activated during purification?

3. Figure 1B: Author mentioned that Salmonella infection triggers caspase-1-dependent pyroptosis in human neutrophils, but the Figure indicates ‘CHA’ but not Salmonella. What is CHA?

4. Y-VAD blocks both caspase-1 and -4. How can the authors be sure that the effect of Y-VAD is not due to blockade of caspase-4?

5. The section on PP34ExoU is very confusing and needs to be re-written (lines 237-246). Am I right to say that ExoU-deficient strains also trigger caspase-1-dependent pyroptosis in neutrophils?

6. PP34ExoU/FliC- is unable to trigger ASC speck in vivo but the lack of inflammasome activation for the double mutant has not been validated in vitro. Does this mean flagellin is required for NLRC4 activation but ExoU suppresses it? Why is that so?

7. Figure 3A – are these capase-1 single KO or Casp1/11 DKO? If they are caspase-1/11 DKO neutrophils, NLRC4 will serve as a better control, since Casp11 is known to trigger NETosis.

8. I am very confused by Figure 3. Line 284 ‘murine BMN with PA01, PA01ExoS-, PP34ExoU or PP34EoxU- strains and measured for the presence of NETs using SEM (Fig. 3A)’. However, I can’t seem to find the condition ‘PP34ExoU-‘ in Figure 3A. Did the authors mean PP34ExoUS142A instead? What is PP34ExoU? Does the strain normally express ExoU or is this introduced into the strain? If they naturally express ExoU, labelling it PP34 will suffice. Since the start of Figure 3 is confusing and poorly labelled, it is challenging to provide any useful comments for the rest of Figure 3.

9. The authors are proposing that neutrophil caspase-1 function is detrimental for host defence when infected with pyroptosis-inducing Pseudomonas, but offers no mechanistic insights into this observation.

10. I recommend to pool data from different experiments to increase the robustness of the study

Minor points

1. Neutrophils are only resistant to caspase-1 but not caspase-11-driven pyroptosis. For clarity and accuracy, this should be changed in the abstract, summary and introduction.

2. I am not sure if the requirement for GSDMD in ROS-driven NETosis still stands, given that a recent article from Dixit’s lab showed that LDC7559 inhibits ROS instead of GSDMD (PMID: 34320407). This should be clarified in the introduction.

3. Reference 30 suggest neutrophil undergo NLRP3 activation and NETosis during SARS-CoV-2 infection, not sterile inflammation.

4. Line 265 typo – Figure 2E not 3E

5. Line 272 and 276 typo – Fig 2F not 2D

6. There are a lot more typo mistakes in the manuscript, please fix them

7. If PP34ExoU is used for most parts of the in vivo characterisation, the ability for them to trigger pyroptosis should be moved to the main figure, not the supplementary.

**Part III – Minor Issues: Editorial and Data Presentation Modifications**

Reviewer #1: Abstract: The authors mention arms race in the abstract but never again- please remove as there is no data supporting a role for this phenomenon in the host microbe arms race.

The data presented do not show that Pad4 gets activated by GSDMD as stated in the abstract, please rephrase

Introduction: Line 149: Data by Rauch et al 2016 suggest Pseudomonas also activates the Naip6-Nlrc4 inflammasome

Statistics: T- test does not seem the appropriate test for several of these experiments, where multiple groups are being compared.

Figure legends: sometimes the authors state in the figure legends: one experiment performed three times. I assume they mean “one experiment representative of three independent experiments” please clarify

Editorial:

Multiple places in manuscript: It is spelled Hoechst dye.

Line 165: I don’t believe WT humans is an appropriate term

205-209: Salmonella infection is referenced in 1B but not shown? Please show the data or remove in the text

214-215 Flagellin is not a toxin please rephrase

220-223: The last panel in 1C suggests, other than discussed in the text, that Flagellin is the sole inflammasome activator as the difference between WT and Casp1 is lost in the FliC- strain. A phenotype in the T3SS system deficient strain can be explained that without the secretion system, no flagellin will leak into the cytosol

246: there is still a significant difference between infected and uninfected Casp1-/- at later timepoints in S1D:, so please rephrase line “in a fully Caspase-1 and GSDMD-dependent manner.” to a more appropriate statement

Line 265: reference to fig3E I believe this reference S2C?

Line 272: refence to 2F instead of 2D

Line 304: “neither PAO1ExoS- nor PP34ExoU- induced Histone release” As far as I can tell there is no ExoU- data shown in these figures, do the authors mean ExoUS142A?

Line 319 “Histone3-Citrullination in a NLRC4-, GSDMD- and CASP-1-dependent manner “- there is still H3-Cit bands visible in S3E in Gsdmd and Nlrc4-knockouts, just weaker, and no Casp1 data is shown, please rephrase ethe statement to reflect that

320: Human blood neutrophils not in S3C but D

370-73: Figures 4D+G: IL-1 levels in the lungs of these mice could just be decreased because of lower bacterial levels, please discuss that in the description of the data.

374-75: The claim about spread not valid based on the data shown as the authors only examine lung tissue but not spread to other organs, please remove.

Figure 1C: please label the X-axis (I assume its minutes)

Figure 3: PAD4 has been shown to be required for NETosis in papers the author cite- so the figure title "Caspase-1-induced neutrophil pyroptosis promotes PAD4-dependent incomplete NETosis" is misleading as it suggests that pAD4 is causing incomplete NETosis, which is not what the authors show

Discussion: Please discuss why PAD4 in this system would lead to only H3 citrunilation but not NETosis as observed by others.

Please discuss the absence of a Caspase-11 phenotype with this bacterium

Reviewer #2: (No Response)

PLOS authors have the option to publish the peer review history of their article (what does this mean?). If published, this will include your full peer review and any attached files.

Reviewer #1: No

Reviewer #2: No
---

## [Decision Letter · Decision Letter 1]

1 Jun 2022

Dear Dr. Meunier,

We are pleased to inform you that your manuscript 'Caspase-1-driven neutrophil pyroptosis and its role in host susceptibility to Pseudomonas aeruginosa' has been provisionally accepted for publication in PLOS Pathogens.

Best regards,

Vincent T Lee

Associate Editor

PLOS Pathogens

David Skurnik

Section Editor

PLOS Pathogens

Kasturi Haldar

Editor-in-Chief

PLOS Pathogens

orcid.org/0000-0001-5065-158X

Michael Malim

Editor-in-Chief

PLOS Pathogens

orcid.org/0000-0002-7699-2064

Congratulations and thank you for thoughtfully addressing the reviewers concerns. There are some minor comments/suggestions from one of the reviewer. Please address these accordingly.

Reviewer Comments (if any, and for reference):

Reviewer's Responses to Questions

**Part I - Summary**

Reviewer #1: I want to thank the authors for addressing my points thoroughly and want to congratulate them on an extensive revision that improved the manuscript significantly.

Reviewer #2: The authors have fully addressed all of my concerns. I congratulate the authors on a very nice piece of work.

However, I do have a few suggestions (text changes) to improve the clarity of the manuscript, listed below.

**Part II – Major Issues: Key Experiments Required for Acceptance**

Reviewer #1: (No Response)

Reviewer #2: N/A

**Part III – Minor Issues: Editorial and Data Presentation Modifications**

Reviewer #1: (No Response)

Reviewer #2: The authors mentioned this is their abstract and introduction 'Notably, deletion of Exotoxins U or S in P. aeruginosa entirely rerouted neutrophil death to Caspase-1-dependent pyroptosis, suggesting that these exotoxins interfere with this pathway.' This statement is inaccurate in my opinion, as PA01 is already capable of inducing pyroptosis, therefore, there is no re-routing. Rather, deletion of ExoU/S increased or enhanced caspase-1-dependent pyroptosis.

The authors also mentioned 'sterile pyroptosis' a couple of times in their results and discussion. I believe sterile inflammation/pyroptosis occurs in the absence of an infection, therefore, I suggest the authors to remove/replace this word.

Line 284-285: These experiments showed that the P. aeruginosa PP34 strain potently induces lysis of Casp1-/- or GsdmD-/- neutrophils (Fig. 1F, S1E Fig.). This sentence is also quite confusing. Do the authors mean ' PP34 induce caspase-1 and gsdmd-independent pyroptosis' instead?

PLOS authors have the option to publish the peer review history of their article (what does this mean?). If published, this will include your full peer review and any attached files.

Reviewer #1: No

Reviewer #2: No

---

## [Editor Report · Acceptance letter]

14 Jul 2022

Dear Dr. Meunier,

We are delighted to inform you that your manuscript, "Caspase-1-driven neutrophil pyroptosis and its role in host susceptibility to Pseudomonas aeruginosa," has been formally accepted for publication in PLOS Pathogens.

Best regards,

Kasturi Haldar

Editor-in-Chief

PLOS Pathogens

orcid.org/0000-0001-5065-158X

Michael Malim

Editor-in-Chief

PLOS Pathogens

orcid.org/0000-0002-7699-2064